# Mi-2β promotes immune evasion in melanoma by activating EZH2 methylation

Cang Li[1,2,15], Zhengyu Wang [ID][3,15], Licheng Yao [ID][4,15], Xingyu Lin[5], Yongping Jian[6], Yujia Li[6], Jie Zhang [ID][7], Jingwei Shao[8], Phuc D. Tran[3], James R. Hagman [ID][9], Meng Cao[10], Yusheng Cong [ID][11], Hong-yu Li[3,16] ✉, Colin R. Goding [ID][12,16] ✉, Zhi-Xiang Xu [ID][6,16] ✉, Xuebin Liao [ID][4,16] ✉, Xiao Miao[13,14,16] ✉ & Rutao Cui [ID][1,16] ✉

Recent development of new immune checkpoint inhibitors has been particularly successfully in cancer treatment, but still the majority patients fail to benefit. Converting resistant tumors to immunotherapy sensitive will provide a significant improvement in patient outcome. Here we identify Mi-2β as a key melanoma-intrinsic effector regulating the adaptive anti-tumor immune response. Studies in genetically engineered mouse melanoma models indicate that loss of Mi-2β rescues the immune response to immunotherapy in vivo. Mechanistically, ATAC-seq analysis shows that Mi-2β controls the accessibility of IFN-γ-stimulated genes (ISGs). Mi-2β binds to EZH2 and promotes K510 methylation of EZH2, subsequently activating the trimethylation of H3K27 to inhibit the transcription of ISGs. Finally, we develop an Mi-2β-targeted inhibitor, Z36-MP5, which reduces Mi-2β ATPase activity and reactivates ISG transcription. Consequently, Z36-MP5 induces a response to immune checkpoint inhibitors in otherwise resistant melanoma models. Our work provides a potential therapeutic strategy to convert immunotherapy resistant melanomas to sensitive ones.

Immunotherapies, especially some new immune checkpoint inhibitors, have been particularly successfully in melanoma. Since 2011, the FDA and EMA have approved four new immunotherapies for patients with the advanced melanoma, including the anti-CTLA-4 antibody ipilimumab (Yervoy), the anti-PD-1 antibodies nivolumab (Opdivo) and pembrolizumab (Keytruda), and the oncolytic virus talimogene laherparepvec (TVEC, Imlygic)[1]. Clinical data shows that 20% of melanoma patients respond to ipilimumab (anti-CTLA-4)[2], 33% respond to

[1]Skin Disease Research Institute, The 2nd Hospital and School of Medicine, Zhejiang University, Hangzhou 310058, China. [2]Research Center for Life Science and Human Health, Binjiang Institute of Zhejiang University, Hangzhou 310053, China. [3]Department of Pharmaceutical Sciences, College of Pharmacy, University of Arkansas for Medical Science, Little Rock, AR 72205, USA. [4]State Key Laboratory of Molecular Oncology, School of Pharmaceutical Sciences, Tsinghua-Peking Center for Life Science, Tsinghua University, Beijing 100084, China. [5]Zhuhai Yu Fan Biotechnologies Co. Ltd, Zhuhai, Guangdong 51900, China. [6]School of Life Sciences, Henan University, Kaifeng 475000, China. [7]National Key Laboratory for Novel Software Technology, Nanjing University, Nanjing, Jiangsu, China. [8]National & Local Joint Engineering Research Center of Targeted and Innovative Therapeutics, International Academy of Targeted Therapeutics and Innovation, College of Pharmacy, Chongqing University of Arts and Sciences, Chongqing 402160, China. [9]Department of Immunology and Genomic Medicine, National Jewish Health, Denver, CO 80206, USA. [10]Affiliated Hospital of Integrated Traditional Chinese and Western Medicine, School of Pharmacy, Nanjing University of Chinese Medicine, Nanjing 210023, China. [11]Key Laboratory of Aging and Cancer Biology of Zhejiang Province, Hangzhou Normal University School of Basic Medical Sciences, Hangzhou 310058, China. [12]Ludwig Institute for Cancer Research, Nuffield Department of Clinical Medicine, University of Oxford, Headington, Oxford OX3 7DQ, UK. [13]Department of Dermatology, Shuguang Hospital of Traditional Chinese Medicine, Shanghai University of Traditional Chinese Medicine, Shanghai 200437, China. [14]The MOE Basic Research and Innovation Center for the Targeted Therapeutics of Solid Tumors, Jiangxi Medical College, Nanchang University, Nanchang, China. [15]These authors contributed equally: Cang Li, Zhengyu Wang, Licheng Yao. [16]These authors jointly supervised this work: Hong-yu Li, Colin R. Goding, Zhi-Xiang Xu, Xuebin Liao, Xiao Miao, Rutao Cui. ✉e-mail: lih1@uthscsa.edu; colin.goding@ludwig.ox.ac.uk; 10140187@vip.henu.edu.cn; liaoxuebin@mail.tsinghua.edu.cn; 0000002623@shutcm.edu.cn; rutaocui@zju.edu.cn

pembrolizumab (anti-PD-1)[3], and 58% respond to a dual immune checkpoint blockade (anti-PD-1+anti-CTLA-4), but with significant toxicity[4,5]. It is critical to note that even though the most responsive cancer patients may maintain long-lasting disease control, one third of those still relapse[6,7].

A successful anti-tumor immune response following PD-1/PD-L1 blockade is believed to require reactivation and proliferation of clones of antigen-experienced T cells in the tumor microenvironment[8,9]. Inadequate anti-tumor T-cell effector function may preclude proper T-cell function to limit the efficacy of immune checkpoint inhibitors[8,10]. Those important factors include high levels of immune suppressive cytokines or chemokines, and recruitment of immune suppressive cells, such as myeloid-derived suppressor cells (MDSCs) and regulatory T cells (Tregs)[9]. Tumor-intrinsic interferon signaling has been demonstrated to control tumor sensitivity to T-cell rejection and subsequently regulates adaptive resistance to immune checkpoint blockades[11,12]. Furthermore, inhibition of p21-activated kinase 4 (PAK4) increased T-cell infiltration and reversed resistance to PD-1 blockade through modulating WNT signaling[13]. *STK11/LKB1* alterations are the prevalent genomic driver for primary resistance to PD-1 inhibitors in *KRAS*-mutant lung adenocarcinoma[14]. In addition, the loss of PTEN decreases T-cell infiltration in tumors to enhance immune resistance and reduce T-cell-mediated cell death[15].

Given the significance of chromatin in modulating gene expression and maintaining genome stability, some chromatin regulatory factors and enzymes are involved in the development of resistance to immunotherapies[16], such as EZH2[17], ARID1A[18] and KDM5B[19]. Specifically, EZH2 inhibition enhances T-cell-targeting immunotherapies in mouse models of melanoma[17,20]. ARID1A, a member of the SWI/SNF family can interact with EZH2 to inhibit IFN-responsive gene expression in cancer cells whose mutations can shape the cancer immune phenotype and immunotherapy[18]. KDM5B, an H3K4 demethylase suppresses anti-tumor immunity by epigenetic silencing of retroelements[19]. The overexpression of PRC2, a multiprotein enzyme complex (EZH2, SUZ12, EED) regulating the trimethylation of lysine 27 on histone H3 (H3K27me3), is present in cancer cells and mediates the repression of IFN-γ-stimulated genes[21]. Furthermore, chromatin-remodeling PBAF contributes to cancer cell immune resistance[22,23] whereas BRG1, a chromatin-remodeling enzyme, has also been implicated in enhancing IFN-stimulated gene transcription[24]. Mutations in other PBAF complex members, such as ARID2 and BRD7, occur in melanoma and can overcome resistance of tumor cells to T-cell-mediated cytotoxicity[23,25].

Mi-2β, also known as CHD4 (chromodomain helicase DNA-binding protein 4), is a CHD family remodeling enzyme in the NuRD complex, which include the histone deacetylases 1 and 2 (HDAC1 and HDAC2), RBBP4/RBBP7, MBD2/MBD3, MTA-1/MTA-2/ MTA-3 and GATAD2A/B[26]. Mi-2β plays important roles in chromatin assembly, genomic stability and gene repression[27]. In this work, we find that Mi-2β plays a key role in regulating adaptive immune response in melanoma. Mi-2β silencing induces the immune response to anti-PD-1 antibody treatment in "cold" melanoma, and the effects are mediated by interferon-stimulated genes (ISGs), such as *Cxcl9* and *Cxcl10*. Mechanistic studies demonstrate that Mi-2β controls the chromatin accessibility of ISGs by binding to EZH2 and promoting its methylation. Methylated-EZH2 subsequently activates the trimethylation of H3K27 to inhibit the transcription of ISGs. Moreover, we develop a specific Mi-2β-targeted inhibitor Z36-MP5. Treatment with Z36-MP5 induces response to anti-PD-1 therapy in "cold" melanoma in vivo.

## Results

### Tumor-intrinsic Mi-2β modulates the resistance of melanoma to T lymphocyte-mediated killing

We focused on identifying tumor-intrinsic epigenetic factors which are crucial in regulating the adaptive immune response in melanoma. To preliminarily identify the key epigenetic factors that regulate cell sensitivity and resistance to T-cell-mediated attack in melanoma, we analyzed the hazard ratio of different epigenetic factors in melanoma with varying levels of T-cell infiltration. Tumor-intrinsic CD8 levels served as a marker to indicate T-cell infiltration[28]. Epigenetic factors were preliminarily recognized as a potential regulator of immune response if their expression level was significantly correlated with the hazard ratio in patients with high CD8 T-cell infiltration only, but not in patients with low CD8 T-cell infiltration. Using these criteria, fifty-five potentially immune-modulatory epigenetic factors were identified (Supplementary Tab. 1). Mouse B16F10 implantation studies were next used to further validate the role of the most correlated genes ($n = 10$) identified in the hazard ratio analysis in regulating T-cell-mediated cytotoxicity. Specifically, each candidate gene was silenced by specific gRNA in B16F10 cells and grafted into C57BL6 mice. B16F10 cells are resistant to immunotherapies, including checkpoint blockade antibodies against PD-1[29,30]. The tumor-infiltrating T cells and B16F10 implantation tumor volume were measured to determine the B16F10 cell response to cytotoxic T cells (Supplementary Fig. 1a, b). *EP400, Mi-2β, PRDM4, NCOA6* or *CARM1* silencing significantly induced the response to T-cell attack in melanoma cells, and led to more than 20% of the melanoma cells to be eliminated by T-cell-mediated killing (Supplementary Fig. 1a). Mi-2β was picked for further analysis due to the epidermal inflammation phenotypes in conditional keratinocyte-specific Mi-2β knockout mouse[31].

Mi-2β is a chromatin-remodeling enzyme with a SNF2-like ATPase domain and plays critical roles in chromatin assembly and genomic stability. To validate the significance of Mi-2β in regulating the immune microenvironment in human melanoma, the correlation between Mi-2β mRNA levels and those of CD8A and CD8B were first analyzed in melanoma patients collected in The Cancer Genome Atlas (TCGA). Mi-2β mRNA levels were negatively correlated with both CD8A and CD8B mRNA expression ($p < 0.01$) (Supplementary Fig. 1c). These results indicate that lower Mi-2β expression correlates with enrichment of CD8+ T-cell infiltration in melanoma. Next, to identify the role of Mi-2β in the immune response in melanoma, the correlation between Mi-2β and GZMB or PRF1 was analyzed. GZMB and PRF1 are crucial for the rapid induction of target cell apoptosis by cytotoxic T lymphocytes (CTL) in cell-mediated immune response[32]. Mi-2β mRNA levels negatively correlated with both GZMB and PRF1 mRNA expression ($p < 0.01$) in melanoma (Supplementary Fig. 1d). These results suggest that expression levels of Mi-2β may be associated with T-cell-mediated killing in melanoma. Consistent with this finding, the repression of Mi-2β expression correlated with a substantial survival benefit in melanoma patients with high CD8+ T-cell infiltration ($p < 0.05$), but not in melanomas with low CD8+ T-cell infiltration (Supplementary Fig. 1e). Collectively, these results suggest a critical role for Mi-2β in regulating melanoma resistance to T-cell-mediated cytotoxicity with tumor-intrinsic Mi-2β levels regulating melanoma sensitivity to T-cell-mediated anti-tumor immunotherapy.

### Mi-2β deficiency enhances the sensitivity to T-cell-mediated killing in melanoma

To identify whether Mi-2β depletion induced an immune response in B16F10 melanoma cells, mouse graft melanomas with shMi-2β virus-infected B16F10 cells were treated using anti-PD-1 antibodies (10 mg/kg) at day 6, 9, 12, 15 and 18 after tumor cell inoculation in immuno-competent C57BL/6 mice. Consistent with previous reports[13,30], mice injected with control B16F10 cells with shScramble were not sensitive to anti-PD-1 treatment. However, Mi-2β silencing combined with anti-PD-1 treatment conferred a substantial inhibition of tumor growth in B16F10 melanoma cells (Fig. 1a, b), and subsequently extended the survival of the treated mice (Fig. 1c). Furthermore, we did not find any changes in cell proliferation and in readily killed function of CD8+ T lymphocytes on B16F10 cells with Mi-2β silencing (Fig. S1F−H). In

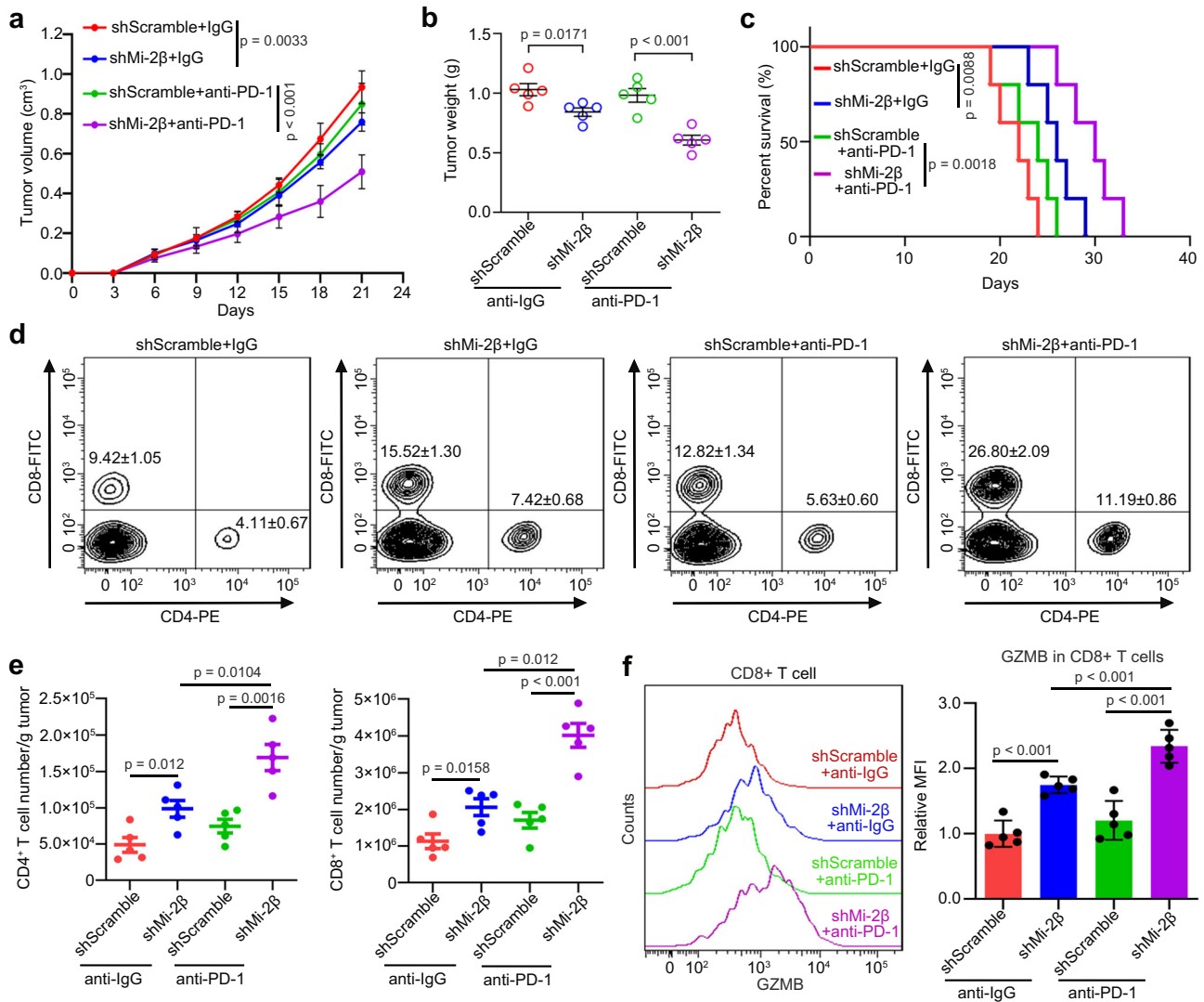

**Fig. 1 | Identification of Mi-2β regulating melanoma cell resistance to immunotherapy. a, b** Mice bearing Mi-2β knockdown or shScramble B16F10 cells were treated with i.p. injection of control IgG (10 mg/kg) or anti-PD-1 (10 mg/kg) antibodies at day 6, 9, 12, 15 and 18 after tumor cell inoculation, tumor volume ($n = 5$) and tumor weight ($n = 5$) were measured. **c** Mouse survival ($n = 5$) over time. Log-rank test was used to determine $P$ value. **d** The representative cell populations of CD4[+] and CD8[+] are shown. **e** Tumor-infiltrating lymphocytes in the graft tumor were measured by flow cytometry. **e** The number of CD4[+] T cells and CD8[+] were gated within CD45[+] T cells ($n = 5$). **f** Granzyme B expression in CD8[+] T was measured and quantified by flow cytometry ($n = 5$). **a, b, e, f** Values represent mean ± SD. The unpaired, two tailed t-test. Source data are provided as a Source Data file.

addition, no alteration was observed in the expression of H-2Db in melanoma cells with or without Mi-2β. Both results exclude that the antigen presentation is upregulated after Mi-2β silencing in melanoma cells. Analysis of the graft tumor microenvironment by flow cytometry showed an increase in CD8[+] and CD4[+] T-cell infiltration detected in the B16F10 tumor graft after Mi-2β silencing, which was strongly augmented by anti-PD-1 treatment (Fig. 1d, e). At the same time, a minor, but non-significant, increase in tumor-infiltrating Treg cells was also detected in the B16F10 tumor graft following Mi-2β silencing, which was not inhibited by anti-PD-1 treatment and/or Mi-2β silencing (Supplementary Fig. 1j). Moreover, a minor to medium increase of GZMB expression and upregulation of activation of CD69, IFN-γ, CD25 and CD107 were detected in tumor-infiltrating CD8[+] T cells from the B16F10 tumor graft after silencing Mi-2β, which was strongly augmented by anti-PD-1 treatment (Fig. 1f and Supplementary Fig. 1k). These data indicate that Mi-2β silencing sensitizes tumor cells and confers a more favorable tumor microenvironment to induce an adaptive immune response to anti-PD-1 treatment in melanoma.

## Loss of Mi-2β induces responses to immunotherapy in BRaf[V600E]/Pten[null] melanoma

To further examine whether Mi-2β depletion induced an adaptive immune response in melanoma in vivo, *Tyr::CreER;BRaf[CA];Pten[lox/lox]* mice were used for the anti-PD-1 antibody treatment. In this mouse strain, induction of Cre-mediated recombination leads to Braf[V600E] expression and Pten inactivation (BRaf[V600E]/Pten[null]) in cutaneous melanocytes, resulting in rapid melanoma initiation and progression[33]. *Mi-2β[lox/lox]* mice[31] were crossed with *Tyr::CreER;BRaf[CA];Pten[lox/lox]* mice to deplete *Mi-*2β in the BRaf[V600E]/Pten[null] melanoma background after tamoxifen injection. Mice with visible melanomas were randomly treated with either control IgG (10 mg/kg) or anti-PD-1 (10 mg/kg) starting at day 9, 12, 15, 18 and 21 after Cre activation (Fig. 2a) and then mouse survival was analyzed. Consistent with previous reports[15], BRaf[V600E]/Pten[null] melanoma is a "cold" tumor, lacking substantial immune infiltration, and was consequently insensitive to anti-PD-1 antibody treatment. There was no significant difference of mouse survival observed on a BRaf[V600E]/Pten[null] background irrespective of

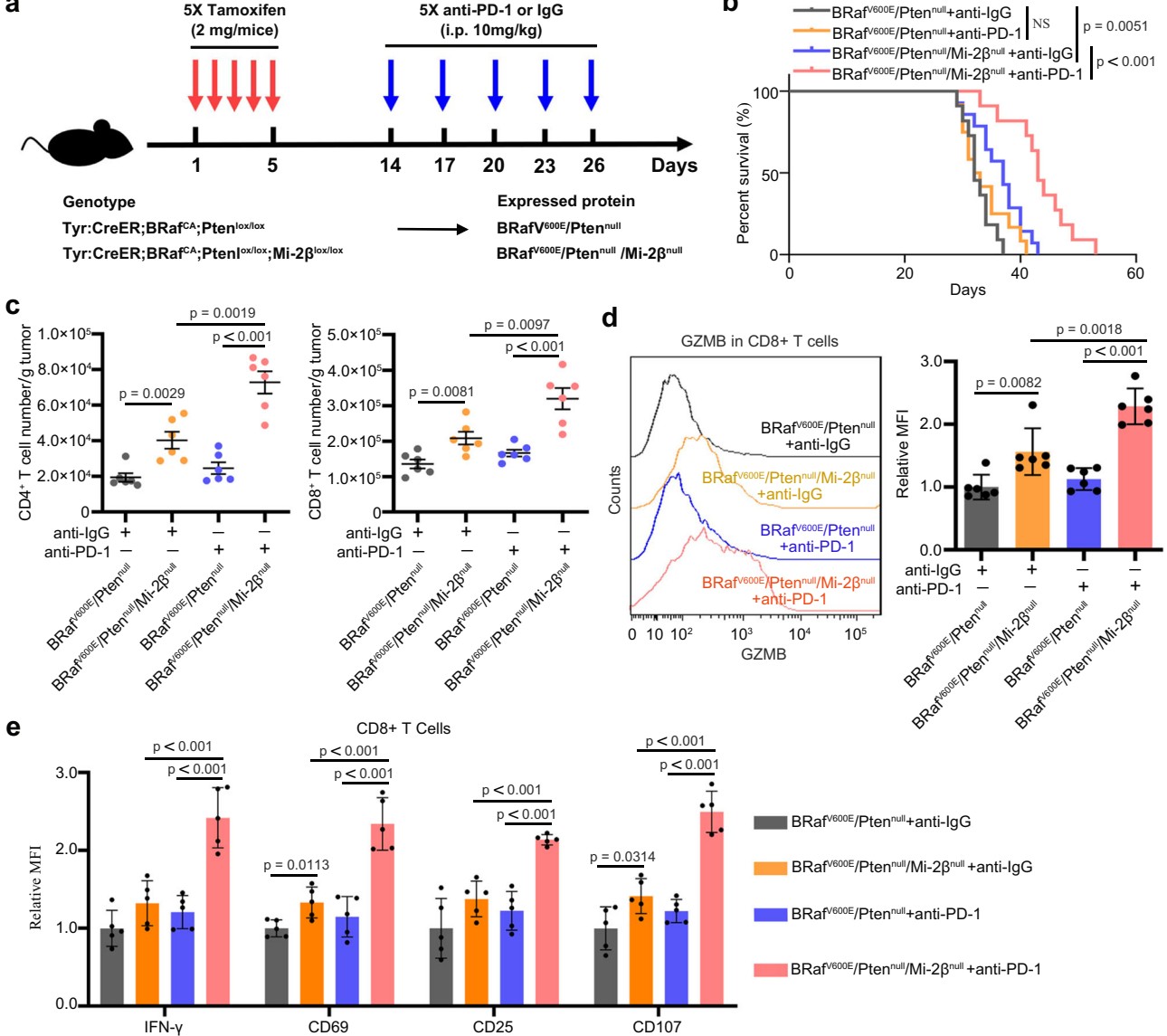

**Fig. 2 | Mi-2β deficiency induced responses to immunotherapy in melanoma.** **a** A schematic for experimental strategy with anti-PD-1 treatment on genetically engineered melanoma mouse model. Mice carrying conditional alleles of *Tyr::CreER;BRaf^CA;Pten^{lox/lox}* or *Tyr::CreER;BRaf^CA;Pten^{lox/lox}Mi-2β^{lox/lox}* were administered with tamoxifen for constant 5 days to activate CreER to cause melanocyte-specific conversion of *Braf* to *Braf^{V600E}*, and the conversion of the *Pten^{lox/lox}* and *Mi-2β^{lox/lox}* alleles to null alleles, which expressed proteins of BRaf^{V600E}/Pten^{null} or BRaf^{V600E}/Pten^{null}/Mi-2β^{null}, respectively. Mice with measurable tumors were randomly treated with either control IgG (10 mg/kg) or anti-PD-1 (10 mg/kg) antibodies at day 9, 12, 15, 18 and 21 after Cre activation. **b** Survival of BRaf^{V600E}/Pten^{null} mice

treated with IgG ($n = 11$) or anti-PD-1 ($n = 12$), and of BRaf^{V600E}/Pten^{null}/Mi-2β^{null} mice treated with IgG ($n = 14$) or anti-PD-1 ($n = 11$). Log-rank test was used for *P* value calculation, NS represents no significance. **c** TILs were assayed with flow cytometry. The number of CD8^+ cells and CD4^+ T cells gated within CD45^+ T cells were demonstrated ($n = 6$). **d** Granzyme B expression in CD8^+ T was determined and quantified by flow cytometry ($n = 6$). **e** Expression of activation markers on CD8^+ T cells were determined by flow cytometry. MFI represents mean fluorescence intensity ($n = 5$). **c**, **d**, **e** Values represent mean ± SD. The unpaired, two tailed t-test. Source data are provided as a Source Data file.

anti-PD-1 treatment (Fig. 2b). IHC staining for the melanoma marker S100 and proliferation marker Ki-67 showed no difference between BRaf^{V600E}/Pten^{null} melanomas with different Mi-2β status (Supplementary Fig. 2a). Intriguingly, treatment of anti-PD-1 significantly extended mouse survival with BRaf^{V600E}/Pten^{null}/Mi-2β^{null} melanoma compared with that of BRaf^{V600E}/Pten^{null} melanoma mice (Fig. 2b).

To further identify whether the Mi-2β knockout-induced anti-PD-1 response correlates with T-cell activation, tumor-infiltrating lymphocytes (TILs) from BRaf^{V600E}/Pten^{null} melanomas with different Mi-2β status were analyzed by flow cytometry (Supplementary Fig. 2b). The populations of infiltrating CD8^+ and CD4^+ T cells were increased to a small extent in the TILs of BRaf^{V600E}/Pten^{null}/Mi-2β^{null} melanoma. This

increase was significantly augmented by the anti-PD-1 treatment (Fig. 2c, d). At the same time, a minor, but non-significant, increase in the Treg population was also detected in BRaf^{V600E}/Pten^{null} melanoma after Mi-2β knockout (Supplementary Fig. 2c). However, anti-PD-1 treatment did not change the Treg cell population in BRaf^{V600E}/Pten^{null} melanomas after Mi-2β knockout (Supplementary Fig. 2c). Moreover, an increase of GZMB expression and upregulation of CD8^+ T-cell activation markers, such as CD69, IFN-γ, CD25 and CD107, were detected in BRaf^{V600E}/Pten^{null}/Mi-2β^{null} melanomas after Mi-2β knockout. These increases were all further strongly augmented by anti-PD-1 treatment (Fig. 2e). To further confirm the finding, melanoma samples collected from different groups were stained for CD8+ T cells by

immunohistochemical staining (IHC). CD8[+] T cells were increased in the TILs in the TME of BRaf[V600E]/Pten[null]/Mi-2β[null] melanomas and the increase was significantly augmented by the treatment of anti-PD-1 antibody (Supplementary Fig. 2d, e). To exclude the possibility that melanoma cells become more sensitive to tumor killing in general in the absence of Mi-2β, BRaf[V600E]/Pten[null] melanoma cells with different Mi-2β status were stimulated with BRaf inhibitor, PLX4032. We found that there was no difference in cell survival in melanoma cells with or without Mi-2β expression upon the treatment of PLX4032. It suggests that melanoma cells did not become more sensitive to tumor killing in general after Mi-2β absence (Supplementary Fig. 2f). Taken together, these results indicate that loss of Mi-2β in melanocytes activates CTLs to induce an anti-PD-1 treatment response in "cold" melanomas in vivo.

## Suppression of Mi-2β upregulates IFN-γ signaling

To determine how Mi-2β shapes the immune response in melanoma, Mi-2β-CRISPR/Cas9-knocked B16F10 cells were used to perform microarray assay. The expression of 1209 genes were significantly repressed (>1.5 fold, $p < 0.05$), and 1283 genes were significantly upregulated (>1.5 fold, $p < 0.05$) after Mi-2β silencing. The deregulated genes identified were further analyzed by Gene Set Enrichment Analysis (GSEA) to identify Mi-2β-regulated gene sets and pathways (Supplementary Tables 2 and 3). Interestingly, IFN-γ signaling was activated after Mi-2β knockout (Supplementary Fig. 3a). IFN-γ production plays a key role in the response to immunotherapy, especially in patients with melanoma[34,35]. Many Mi-2β-controlled IFN-γ-responsive genes, such as *Cxcl9, Cxcl10, CD74, Irf1, and CD40*, function in T-cell chemoattraction, antigen presentation, and T-cell targeting and activation. Specifically, expression of cytokines such as *Cxcl9, Cxcl10, Cxcl11* and *Ccl5* were upregulated after Mi-2β silencing (Supplementary Fig. 3b). These cytokines play a key role in inducing and recruiting effector T cells expressing the CXCR3 chemokine receptor into tumor microenvironment to induce anti-tumor immunity[34–36]. Several antigen presentation genes, such as *Tap1* and *CD74* and some regulators involved in tumor cell immunogenicity, such as *Irf1, Icam1* and *CD40* were also upregulated by Mi-2β knockout in vitro (Supplementary Fig. 3b).

To confirm the regulation of Mi-2β on the downstream targets from IFN-γ signaling, the expression of ISGs in the IFN-γ pathway were measured in Mi-2β-depleted B16F10 cells. Mi-2β silencing significantly upregulated the mRNA expression of *Cxcl9, Cxcl10, Cxcl11, Ccl5, Tap1, CD74, Irf1, Icam1, CD40, Fas* and *PD-L1* (Supplementary Fig. 3c) and enhanced the paracrine secretion of *Cxcl9* and *Cxcl10* (Fig. 3a). In vivo, TIMER analysis[37] indicated that Mi-2β mRNA levels negatively correlated with CCL5, CD74 and CD40 mRNA levels in patients in the TCGA melanoma cohort ($p < 0.01$) (Supplementary Fig. 4a). There was a negative correlation tendency between Mi-2β and *Cxcl9* and *Cxcl10* at the mRNA level in melanomas in TCGA. However, the correlation did not reach the statistical significance. These data indicate that the Mi-2β-regulated immune response is mediated, at least in part, by IFN-γ signaling pathways in melanoma. To identify how Mi-2β impacts the responses to anti-PD-1 treatment, the expression levels of Cxcl9 and Cxcl10 were measured by ELISA in melanomas collected from mice shown in Fig. 1b. Upregulation of Cxcl9 and Cxcl10 was detected after Mi-2β silencing and anti-PD-1 treatment in melanomas (Fig. 3b). In addition, we also measured these factors in the BRaf[V600E]/Pten[null] melanomas collected in Fig. 2b. Upregulations of *Cxcl9, Cxcl10, Cxcl11, Ccl5, Tap1, CD74, Irf1, Icam1, CD40, Fas* and *PD-L1* were detected after Mi-2β silencing and the anti-PD-1 treatment in BRaf[V600E]/Pten[null] melanomas (Supplementary Fig. 4b). To validate that Mi-2β-regulated immune response is based on the enhanced IFNγ-signaling, we detected CXCR3 in the TME of Mi-2β-deficient melanomas by IHC with a specific anti-CXCR3 antibody. Consistently, CXCR3 was upregulated in the TME of Mi-2β-deficient melanomas (Supplementary Fig. 4c, d).

The genomic Mi-2β localization is highly enriched at transcription start sites where it plays an important role in transcriptional repression[38]. To investigate the molecular mechanisms underlying Mi-2β-mediated repression of IFN-γ signaling, ATAC-seq was employed to identify the status of global chromatin accessibility. Specifically, 2980 upregulated differentially accessible regions (DARs) and 3049 down-regulated DARs were identified in B16F10 cells with Mi-2β silencing (Supplementary Fig. 5). These DARs were predominantly located in the intergenic, intronic, and promoter regions (Fig. 3c). The changes of the ATAC-seq peaks within loci were associated with the IFN-γ signals (Fig. 3d). To validate the finding in ATAC-seq, we performed a chromatin immunoprecipitation (ChIP) assay using a specific anti-Mi-2β antibody with anti-Stat1 serving as a positive control. The results indicate that Mi-2β bound to the promoters of *Cxcl9* and *Cxcl10* (Fig. 3e). Collectively, these data indicate that loss of Mi-2β remodels chromatin accessibility and facilitates the exposure of the regulatory regions of ISGs leading to their transcriptional activation.

## Mi-2β promotes the methylation of EZH2 to inhibit the transcription of interferon-stimulated genes

To further identify how Mi-2β regulates ISGs expression, liquid chromatography-tandem mass spectrometry (LC-MS/MS) was performed to identify co-factors of Mi-2β protein. The total number of unique peptides were used to evaluate which proteins are potential co-factors (Supplementary data 1). Interestingly, enhancer of zeste homolog 2 (EZH2) was identified as one of the abundant chromatin regulatory proteins associated with the Mi-2β protein complex (Fig. 4a).

EZH2 is a histone-lysine N-methyltransferase enzyme and is the functional enzymatic component of the polycomb repressive complex 2 (PRC2). EZH2 functions to methylate Lys-27 on histone 3 (H3K27me) by transferring a methyl group from the cofactor S-adenosyl-L-methionine (SAM), which is associated with long term transcription repression[39]. To confirm the interaction between Mi-2β and EZH2 proteins, reciprocal co-immunoprecipitation was performed using different melanoma cells expressing endogenously or exogenously Mi-2β and/or EZH2 proteins. Co-IPs indicate that EZH2 and Mi-2β interacted each other in melanocytes (Fig. 4b and Supplementary Fig. 6a). To identify whether the interaction between EZH2 and Mi-2β functions in regulating the downstream targets of Mi-2β signaling, the expression of ISGs, including *Cxcl9* and *Cxcl10*, were measured in EZH2-depleted B16F10 and mouse primary melanoma cells. EZH2 silencing enhanced the paracrine secretion of Cxcl9 and Cxcl10 (Fig. 4c) and significantly upregulated the mRNA expression of *Cxcl9* and *Cxcl10* (Supplementary Fig. 6b, c). Previous reports have demonstrated that the trimethylation of lysine 27 on histone H3 (H3K27me3) mediates the EZH2-repressed the expression of ISGs[40]. We confirmed that H3K27me3 activation was significantly repressed after EZH2 silencing in vitro (Fig. 4d) and found that H3K27me3 activation mediated the Mi-2β/EZH2 complex-regulated expression of ISGs (Supplementary Fig. 6d, e). These results were further confirmed in melanoma cells with EZH2 silencing[41] (Supplementary Fig. 6f, g). We further confirmed that H3K27me3 levels at the *Cxcl9* and *Cxcl10* promoter region were repressed after Mi-2β silencing (Fig. 4e). These observations indicate that the interactions between Mi-2β protein and EZH2 protein regulate the activation of ISG transcription, which is mediated by H3K27me3.

To identify the interaction domain between Mi-2β and EZH2 proteins, a series truncated EZH2 constructs were amplified (Fig. 4f and Supplementary Fig. 6h). Co-IPs indicated that Mi-2β protein binds to the CXC domain of EZH2 (Fig. 4g). To further identify how Mi-2β regulates epigenetic modification of H3, we detected the post-translational modifications of EZH2, especially after its interaction with Mi-2β. Monomethylation modification of EZH2 was detected in EZH2/Mi-2β complex (Fig. 4h). Previous studies have demonstrated

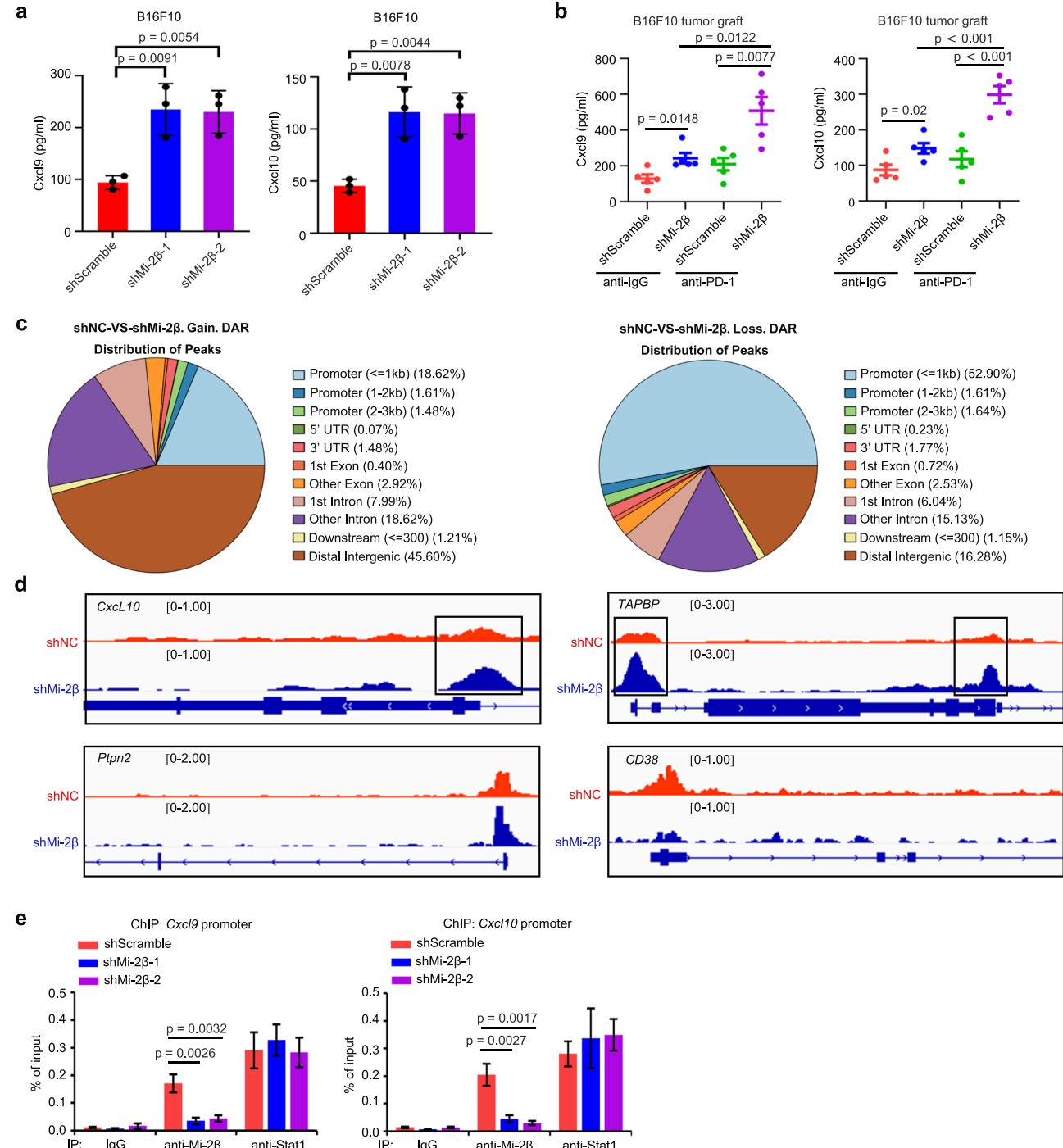

**Fig. 3 | Depletion of Mi-2β promotes the transcription of interferon-stimulated genes by promoting chromatin remodeling. a** The amount of secreted Cxcl9 or Cxcl10 were measured in B16F10 cells with Mi-2β silencing by ELISA (*n* = 3). **b** The graft melanomas were isolated and cultured in PBS with the same amount of cells for 4 h (for each group graft melanomas = 5), and then the secreted amount of the chemokines Cxcl9 and Cxcl10 in the culture medium was measured by ELISA. **c** Genomic distribution of DARs and binding consensus of the TFs. **d** ATAC-seq bedgraph panels of the gene locus showing the peak locations relative to the TSS. The panels were compared with ATAC signals between scramble and Mi-2β knockdown B16F10 cells. **e** ChIP assays were performed to detect Mi-2β binding on the promoter of *Cxcl9* and *Cxcl10* genes in both shScramble and Mi-2β knockdown B16F10 cells, with IP by anti-Stat1 was used as the positive binding control (*n* = 3). **a**, **b**, **e** Values represent mean ± SD. The unpaired, two tailed t-test. Source data are provided as a Source Data file.

that methylated-EZH2 promotes its own enzyme activity[42]. EZH2 automethylation allows PRC2 to modulate its histone methyltransferase activity by sensing histone H3 tails, SAM concentration, and perhaps other effectors[42]. Thus, we postulated that Mi-2β promotes the trimethylation of H3K27 by promoting methylation of EZH2. To verify this hypothesis, mass spectrometry was performed to detect EZH2 methylation sites. We found that EZH2 exhibited monomethylation at K510 and K735 (Supplementary Fig. 6i). EZH2 point mutation experiments indicated that the trimethylation of H3K27 was significantly reduced after exogenously expressing an EZH2 K510A mutant in melanoma cell with stable EZH2 silencing (Supplementary Fig. 6j). This result indicates that methylation of EZH2 at K510 is crucial in the

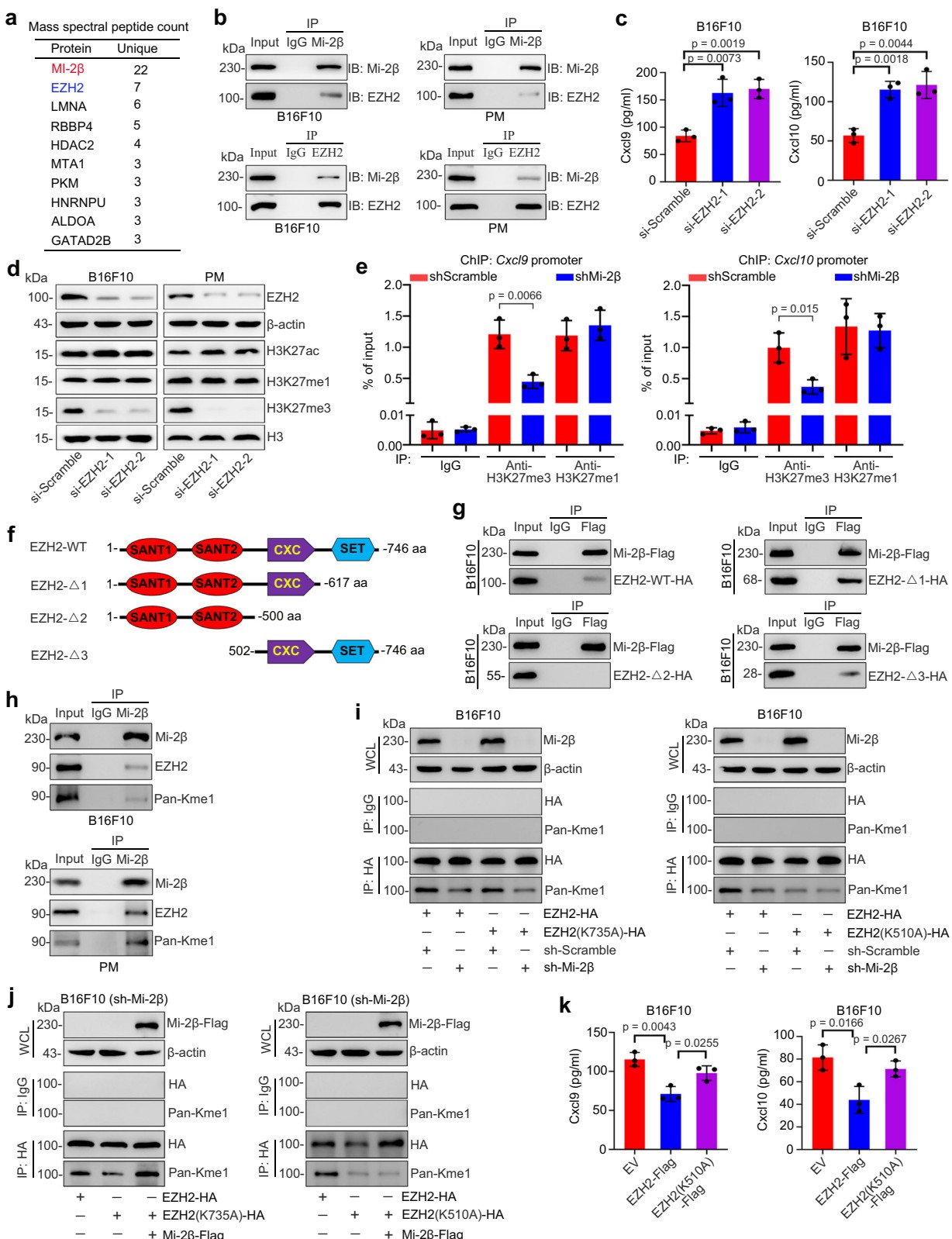

activation of the methylation of H3K27. Furthermore, we found that the K510 EZH2 methylation was Mi-2β dependent (Fig. 4i, j and Supplementary Fig. 6k). Consistent with this, paracrine Cxcl9 and Cxcl10 was significantly upregulated after the introduction of K510A mutant EZH2 into melanoma cells with endogenous EZH2 silencing (Fig. 4k). Taken together, these results indicate that Mi-2β binds to the CXC domain of EZH2 to promote EZH2 K510 methylation. The methylated-

EZH2 subsequently activates the trimethylation of H3K27 to inhibit the ISG transcription.

## Screening and identification of Mi-2β inhibitors

Given the pivotal role of Mi-2β in regulating the immune response, targeting Mi-2β would represent a potential therapeutic strategy in melanoma immunotherapy, especially in combination with anti-PD-1

**Fig. 4 | Mi-2β promotes the K510 methylation of EZH2 to inhibit the transcription of ISGs. a** Mass spectral peptide count of Mi-2β-interacting proteins. **b** Endogenous Mi-2β-EZH2 interactions were detected by immunoprecipitation in B16F10 and the primary mouse melanoma cells (PM) ($n = 3$, independent experiments). **c** The amount of secreted Cxcl10 was measured in B16F10 cells with EZH2 silencing by ELISA ($n = 3$). **d** The epigenetic modifications of H3 (H3K27ac, H3K27me1 and H3K27me3) were measured in B16F10 cells or PM cells with EZH2 silencing by Western blot ($n = 3$, independent experiments). **e** ChIP–quantitative polymerase chain reaction for enrichment of H3K27me3/H3K27me1 at gene locus in B16F10 cells with shMi-2β ($n = 3$). **f, g** Mapping the interaction interface of EZH2 with Mi-2β by immunoprecipitation. HA-tagged EZH2 WT or deletion mutants and Flag-tagged Mi-2β were used as indicated and detected by immunoprecipitation in B16F10 cell. SANT1, SANT domain I; SANT2, SANT domain II; CXC, cysteine-rich domain; SET, methyltransferase catalytic domain (g: $n = 3$, independent experiments). **h** EZH2 lysine methylation was detected by immunoprecipitation with specific anti-Mi-2β antibodies, anti-EZH2 antibodies or anti- pan-monomethylated lysine antibodies, respectively ($n = 3$, independent experiments). Mi-2β (**i**) levels or total Mi-2β-Flag (**j**) in the whole-cell lysate (WCL) and pan-methylated lysine levels of immunoprecipitated EZH2, EZH2 K735A or EZH2 K510A were detected in B16F10 melanoma cells. Immunoglobulin G (IgG) Immunoprecipitation served as a negative control ($n = 3$, independent experiments). **k** The amount of secreted Cxcl9 and Cxcl10 were measured by ELISA in B16F10 cells of stable EZH2 silencing and EZH2 WT or mutant EZH2 reintroduction ($n = 3$). **c, e, k** Values represent mean ± SD. The unpaired, two tailed $t$-test. Source data are provided as a Source Data file.

antibody therapies. To screen small molecules that inhibit Mi-2β activity, Homology Modeling was carried out using Structure Prediction Wizard in Prime[43,44]. Mi-2β belongs to the CHD family of chromatin remodelers, which share highly conserved ATPase/helicase domains[45,46]. The homology model of Mi-2β was generated using the yeast CHD1 structure (PDB code: 3MWY) as a template and the sequence obtained from Uniprot[47], which clearly depicted the interaction between Mi-2β binding pocket and ATP (Supplementary Fig. 7a). Virtual screening was done with the enzyme hinge region ligands database and nucleoside mimetic database from Enamine. All ligands of ~23,010 compounds were docked to the ATP-binding site using SP docking and postprocessed with Prime MM-GBSA. Ligands with a methyldihydroimidazopyridinone structure were predicted to bind best to the ATP-binding region of Mi-2β. To analyze biochemically the inhibitory activity of those inhibitors, a Fluorescence Resonance Energy Transfer (FRET)-based nucleosome repositioning assay was designed[48,49] and modified using recombinant purified human Mi-2β protein to screen an in-house library of small molecular compounds with a methyldihydroimidazopyridinone structure (Fig. 5a). Briefly, the recombinant nucleosome substrates consist of a Cy5-labeled human histone octamer (H2A T120C-Cy5) wrapped with 5′ Cy3-labeled DNA, which contains a terminal nucleosome 601 positioning sequence. The 601 sequence provides the most thermodynamically preferred locations on DNA for a histone octamer. FRET signaling was monitored by exciting the nucleosomes at the Cy3 absorption maximum and measuring Cy5 emissions and consequently the FRET signal is at a maximum at the assembled starting point. In the presence of ATP, Mi-2β induces the histone octamer to translocate along the DNA such that the Cy3-labeled DNA 5′ end is moved away from the Cy5-labeled octamer and consequently the FRET signal is decreased (Fig. 5a). The reaction conditions for nucleosome repositioning were modified through multiple rounds of optimization and validation (Supplementary Fig. 7b, c). Z36 was initially identified as the best hit with IC$_{50}$ values of $6.971 \pm 2.072\,\mu M$ (Supplementary Fig. 7d). Structure-Activity Relationship (SAR) studies were further used to improve the specificity and efficacy of Z36 for Mi-2β inhibition. Through iterative rounds of structure-activity optimization and in vitro assay screens, Z36-MP5 was found to have a high inhibitory activity on Mi-2β function where it was predicted to dock into the ATP-binding pocket of Mi-2β (Fig. 5b, c), with its methyl group extended to a solvent-exposed channel lined with the side chains of Tyr729, Leu755, Met966, and Ile1163. Z36-MP5 could generate H-bonds with Mi-2β via the O atom of its keto group with His727, the O atom of amide group with Gly756, and protonated N atom of imidazole group with Asp873. In vitro assays indicated that Z36-MP5 had IC$_{50}$ values of $0.082 \pm 0.013\,\mu M$ against Mi-2β (Fig. 5d), ~85-fold more potent than the original compound Z36. Moreover, an ATP acyl phosphate probe assay[50] was performed by ActivX Biosciences inc. To profile of Z36-MP5 inhibition on ATPases in native cell lysates, in which the protein-protein interactions remained intact. Z36-MP5 showed less than 35% inhibition at a concentration of $1\,\mu M$ against a panel of 233 diverse ATPases (Supplementary data 2). To further evaluate the specificity of Z36-MP5, a KINOMEscan screening with 468 kinases in the platform at DiscoveRx (San Diego, CA) was conducted (Supplementary Fig. 7e). We also found that the inhibitory effects of Z36-MP5 was the histidine 727 of Mi-2β dependent (Supplementary Fig. 7f). Z36-MP5 was not significantly active on any targets tested. These results suggest that Z36-MP5 has a high Mi-2β ATPase selectivity and specificity.

**Z36-MP5 is a potent and effective inhibitor for Mi-2β and stimulates T-cell-mediated cytotoxicity**

Z36-MP5 was chosen for further validation and experimental therapeutics in vivo. The IC$_{50}$ of Z36-MP5 against Mi-2β was increased with increasing concentration of ATP ($10–300\,\mu M$) (Fig. 5e), suggesting that Z36-MP5 functions as an ATP-competitive inhibitor. To investigate its cellular inhibitory activity, B16F10 cells were treated with Z36-MP5 at concentrations ranging from 5 to $100\,\mu M$, and the activation of Mi-2β target genes measured by RT-qPCR. Z36-MP5 at $25\,\mu M$ induced Mi-2β target gene expression including *Cxcl9* and *Cxcl10* (Fig. 5f) in B16F10 cells. Importantly, monitoring mouse weight (Supplementary Fig. 8a) and organ tissue histological staining (Supplementary Fig. 8b) showed Z36-MP5 treatment was tolerated without significant toxicity in C57BL/6 mice. In addition, the pharmacokinetic properties of Z36-MP5 in Sprague-Dawley rats with administration of intraperitoneal injection dose of 1.0 mg/kg. The results showed that Z36-MP5 exhibited decent pharmacokinetic parameters with a half-life $T_{1/2}$ of 0.45 h and $C_{max}$ of 3.96 μg/mL (Supplementary Fig. 8c). The proliferation of tumor cells and the killing potential of T cells were not significantly inhibited after Z36-MP5 treatment (Supplementary Fig. 8d, e). To exclude any effects on the host cells, we implanted Mi-2β-null melanoma cells into C57BL6 mice and then treated the mice with Z36-MP5 accompanied by anti-PD-1 therapy. No detectable inhibitory effect of Z36-MP5 was observed on tumors generated from Mi-2β-null cells (Supplementary Fig. 8f). This result indicates that the therapeutic outcome of Z36-MP5 is dependent of Mi-2β in melanoma cells. We also found that the treatment effect of Mi-2β inhibition was EZH2 dependent (Supplementary Fig. 8g). Altogether, these data indicate that Z36-MP5 is a potent and effective inhibitor for Mi-2β.

To determine whether Z36-MP5 represented a potential therapeutic option for melanoma immunotherapy, especially in combination with immunotherapy in vivo, syngeneic mouse melanoma developed by subcutaneously grafted B16F10 in C57BL/6 mice were randomly treated with Z36-MP5 (30 mg/kg) and/or anti-PD-1 (10 mg/kg). The results showed that the combinational treatment of Z36-MP5 and anti-PD-1 conferred a substantial inhibition on tumor growth (Fig. 6a, b) and extended mouse survival (Fig. 6c) compared with control treatment. Z36-MP5 treatment alone induced a moderate increase in the CD8$^+$ T-cell TILs in graft melanomas that was augmented by combining with anti-PD-1 therapy (Fig. 6d, e). However, the populations of CD4$^+$ T-cell and Treg cells were not changed significantly by either the individual or combinational treatments (Supplementary Fig. 9a). An upregulation of GZMB expression in

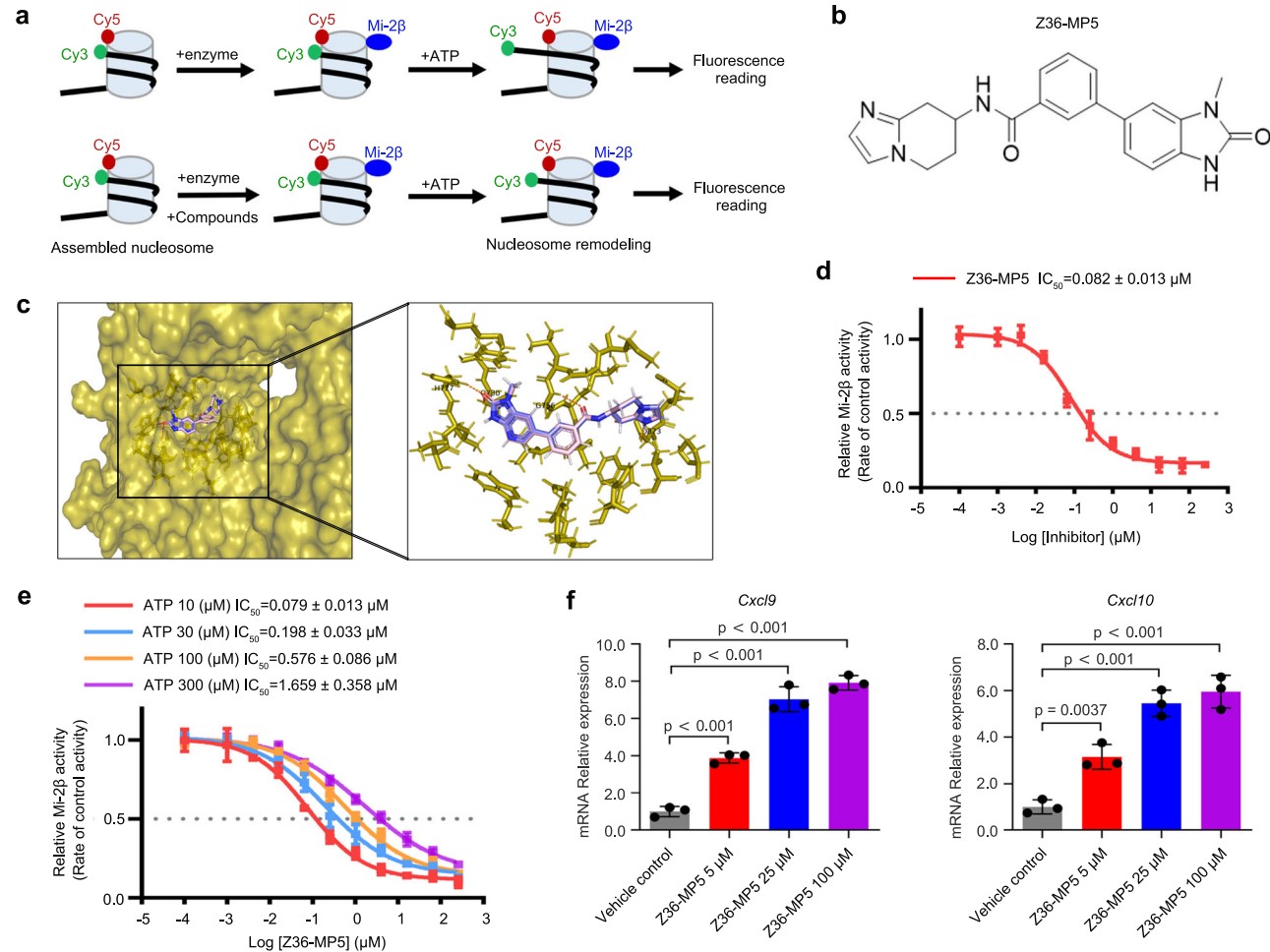

**Fig. 5 | Screening and identifying Mi-2β inhibitors. a** Schematic representing in vitro screen assay for testing Mi-2β chromatin modulatory activity using FRET-based nucleosome repositioning assay. **b** The chemical structure of Z36-MP5. **c** Orientation of Z36-MP5 to homologized Mi-2β. Z36-MP5 was docked into the ATP-binding pocket of homologized Mi-2β. The methyl group of Z36-MP5 extended to a solvent-exposed channel lined with the side chains of Tyr729, Leu755, Met966, and Ile1163, with generating H-bonds via the O atom of keto group with His727, O atom of amide group with Gly756, and protonated N atom of imidazole group with Asp873. The atoms of Z36-MP5 were colored as follows: carbon pink, oxygen red, nitrogen blue, and hydrogen white. The H-bonds between Z36-MP5 and homologized Mi-2β were shown as light-yellow dashed lines. **d** The inhibitory activity of Z36-MP5 for Mi-2β chromatin modulatory activity, measured as fold change of Mi-2β activity treated with control vehicle ($n = 3$). **e** The inhibitory activity of Z36-MP5 with $IC_{50}$ values against Mi-2β at different ATP concentrations ($n = 3$). **f** The expression of *Cxcl9* and *Cxcl10* mRNA in B16F10 cells treated with Z36-MP5 as indicated concentration for 24 h was determined with RT-qPCR assay ($n = 3$). **f** Values represent mean ± SD. The unpaired, two tailed t-test. Source data are provided as a Source Data file.

tumor-infiltrating CD8+ T cells was detected in tumors treated with Z36-MP5, as well as the activation markers CD69, IFN-γ, CD25 and CD107, whose expression was augmented by combinatorial treatment with anti-PD-1 (Fig. 6f and Supplementary Fig. 9b).

The potential of Z36-MP5 therapy was further tested in the *Tyr::CreER;BRaf^{CA};Pten^{lox/lox}* mouse melanoma model. After tamoxifen administration, mice with visible melanomas were randomly treated with Z36-MP5 (30 mg/kg) once a day starting at day 9 and/or anti-PD-1 (10 mg/kg) five times at day 9, 12, 15, 18 and 21 after Cre activation. Z36-MP5 in combination with the anti-PD-1 antibody treatment significantly extended mouse survival in the BRaf^{V600E}/Pten^{null} melanoma mice (Fig. 6g). However, anti-PD-1 treatment alone did not extend mouse lifespan in the BRaf^{V600E}/Pten^{null} mice, consistent with the previous reports that BRaf^{V600E}/Pten^{null} melanoma are insensitive to anti-PD-1 treatment[15] (Fig. 6g). To identify the role of Z36-MP5 treatment in regulating the tumor immune microenvironment, TILs were assayed by flow cytometry. Z36-MP5 treatment alone moderately induced the CD8+ T-cell population, which was further augmented by anti-PD-1

treatment (Fig. 6h). However, the CD4+ T-cell and Treg populations in BRaf^{V600E}/Pten^{null} mouse melanomas were not affected by either Z36-MP5 alone or in combination with anti-PD-1 treatment in BRaf^{V600E}/Pten^{null} melanoma (Supplementary Fig. 9c). Increased expression of GZMB, CD69, IFN-γ, CD25 or CD107 in CD8+ T cells was detected in BRaf^{V600E}/Pten^{null} melanomas, and their induction was further augmented by the anti-PD-1 treatment (Fig. 6i and Supplementary Fig. 9d). To identify whether Z36-MP5-induced melanoma growth inhibition is CD8 T-cell dependent, the treatment effect of Z36-MP5 was evaluated in mice with CD8 T cells neutrilzation[51]. The neutrilzation of CD8 T cells completely abolished the effectiveness of Z36-MP5, resulting in rapid tumor growth (Fig. 6j–l). These results indicate that Z36-MP5 represents an effective combinational immunotherapeutic option of anti-PD-1 treatment in melanoma.

To further evaluate the translational potential of Z36-MP5, we compared the experimental therapeutic effects between Z36-MP5 and the EZH2 inhibitors (GSK126), which have been FDA approved for patients with epithelioid sarcoma[52] and are in Phase I/II clinical trials for

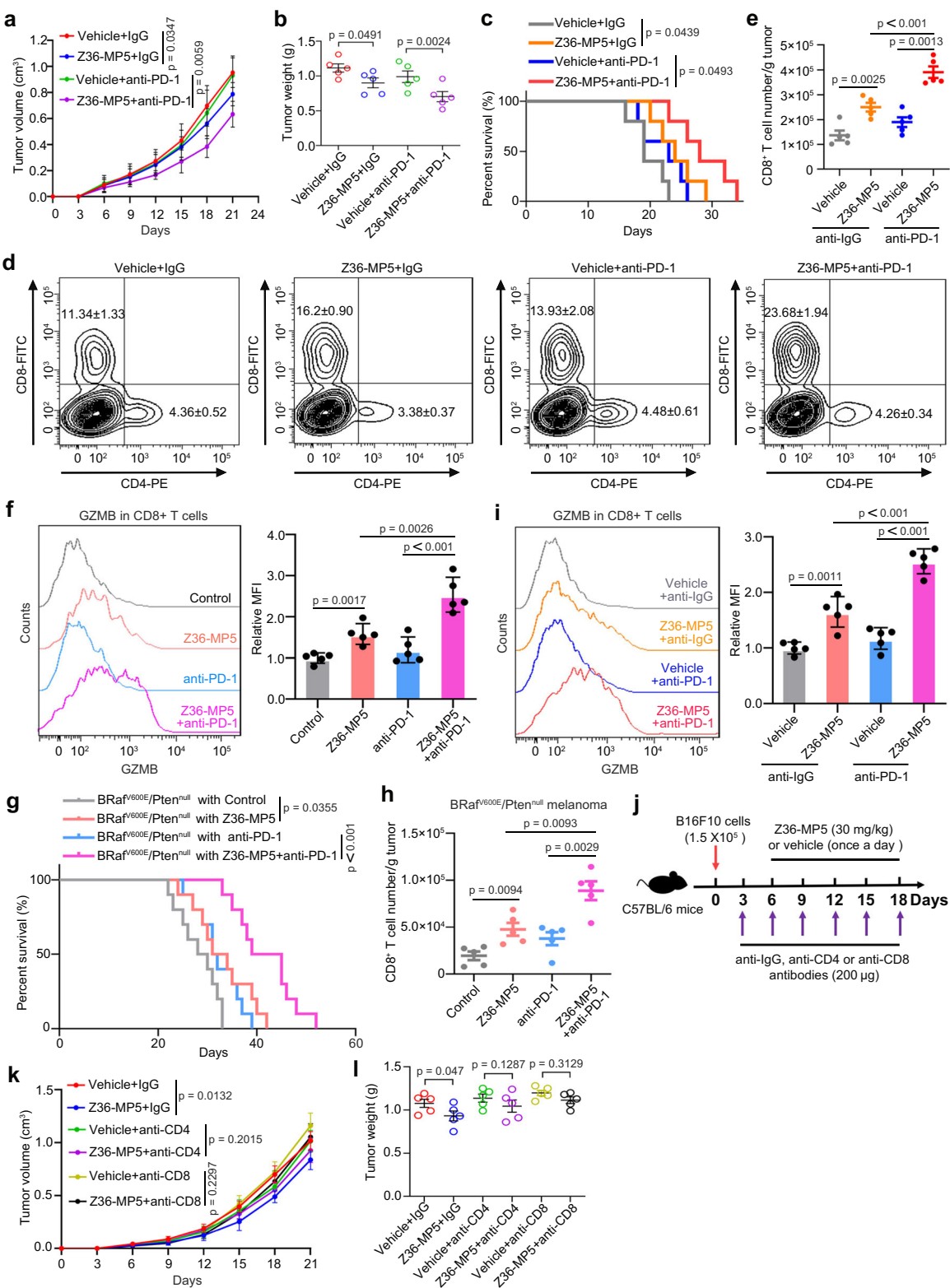

patients with metastatic melanoma. Combination therapies between anti-PD-1 antibodies and Z36-MP5 or EZH2 inhibitors were tested in C57BL/6 mice with B16F10 cells inoculation. The combination therapy between Z36-MP5 and anti-PD-1 antibodies extended the survival of B16F10 implantation mice more than the combination therapy between GSK126 and anti-PD-1 antibodies (Supplementary Fig. 10a). To identify the underlying molecular basis whereby Z36-MP5 has better

effects than GSK126, Z36-MP5 or GSK126-induced H3K27me3 and H3K27Ac were detected in melanoma cells. Previous reports have demonstrated that the EZH2 inhibitors repressed the level of H3K27me3 and at the same time activated H3K27ac levels and subsequently abated the activation of ISGs and the effects of cancer immunotherapy[53]. We confirmed these results, and found that Z36-MP5 repressed the level of H3K27me3 in melanoma cells. However,

**Fig. 6 | Mi-2β inhibitor Z36-MP5 induced immune response to immunotherapy in resistant melanoma. a** Mice bearing B16F10 cell graft were treated with control IgG or anti-PD-1 antibodies, and vehicle control or Z36-MP5, as indicated, and the growth of tumor grafts was shown. For each group n = 5. Mice bearing B16F10 cells were treated with control IgG or anti-PD-1 antibody, and vehicle control or Z36-MP5, as indicated. For each group n = 5. Tumor weight (**b**) and mouse survival curve (**c**) were shown, with log-rank test for mouse survival curve P value. **d**, **e** Tumor-infiltrating lymphocytes (TILs) were assayed and quantified for CD4$^+$ and CD8$^+$ T-cell number in total CD45$^+$ cells by flow cytometry (n = 5). **f** Granzyme B expression in CD8$^+$ T was determined and quantified by flow cytometry (n = 5). **g** Mice carrying conditional alleles of *Tyr::CreER;BRaf^{CA};Pten^{lox/lox}* were administered with tamoxifen for 5 days to activate CreER to cause melanocyte-specific conversion of *Braf^{CA}* to *Braf^{V600E}*, and the conversion of the *Pten^{lox/lox}* alleles to null alleles, which express proteins of BRaf$^{V600E}$/Pten$^{nul}$. Mice with measurable tumors were randomly treated with either control IgG (10 mg/kg) or anti-PD-1 antibodies (10 mg/kg) and Z36-MP5 (30 mg/kg/day) as indicated. For each group n = 10. Mouse survival was shown with log-rank test for P value. **h** TILs were assayed by flow cytometry. The number of CD8$^+$ T cells gated within CD45$^+$ T cells were demonstrated (n = 5). **i** The expression of Granzyme B in CD8$^+$ T was determined and quantified by flow cytometry (n = 5). MFI, mean fluorescence intensity. **j** A schematic for experimental strategy with anti-CD8 antibodies and Z36-MP5 treatment on melanoma mouse model. **k**, **l** Mice bearing B16F10 cell graft were treated with control IgG, anti-CD4 or anti-CD8, and then treated with Z36-MP5. The tumor volum (**k**) and weight (**l**) were shown. For each group n = 5. **a**, **b**, **e**, **f**, **h**, **i**, **k**, **l** Values represent mean ± SD. The unpaired, two tailed t-test. Source data are provided as a Source Data file.

compared to GSK126 treatment, the H3K27ac level only slightly increased in PM cells after treatment with Z36-MP5. Consistently, ISGs were more efficiently induced after Z36-MP5 treatment than GSK126 or MS177 treatment in melanoma cells (Supplementary Fig. 10b–d). These data indicate that Z36-MP5 treatment confers a more favorable tumor microenvironment to cytotoxic CD8$^+$ T cells for overcoming the resistance of melanoma to immunotherapy. The ability to target Mi-2β and recover ISG and inflammatory signals by Z36-MP5 might be further evaluated for integration of combinational immunotherapy in patients with melanoma and other immune resistant cancers in future translational and clinical research.

## Discussion

Given heterogeneity of cancer cells and dynamic evolvement of tumor microenvironment, identifying and demonstrating the potential regulatory factors, which target and modulate antigen presentation and interferon signaling pathways, will be promising in enhancing response, or resistance recovery in cancer immunotherapy. Tumor cell-intrinsic resistance mechanisms of immunotherapies are deeply explored and identified, including processing and presentation of neoantigen by the major histocompatibility complex (MHC)[54,55] and absence of pre-existing T-cell infiltration caused by a lack of T-cell-recognized antigens[56] or MHC[8]. Melanoma-intrinsic Wnt pathway was demonstrated to contribute to a lack of T-cell-melanoma recognition to prevent anti-tumor immunity[57]. The alteration of antigen presentation also regulates the interaction and recognition of tumor cell and T-cell recognition, and of interferon signaling pathways to induce resistance of immunotherapy[6].

More and more studies have demonstrated that some chromatin regulatory factors are crucial in regulating resistance to anti-PD-1 antibodies in melanoma[16], such as EZH2[17], ARID1A[18] and KDM5B[19]. Here Mi-2β, a chromatin-remodeling enzyme, was identified to regulate resistance to T-cell-mediated cytotoxicity and immunotherapy (Fig. 2b, c). The human Mi-2 protein was discovered as an autoantigen in dermatomyositis in 1995[58]. The most well-studied function of Mi-2/NuRD is its indispensable role in cardiac muscle cell identity[59] and haematopoietic development, including T and B lymphocytes[60–62]. The conditional knockout of Mi-2β in mouse keratinocytes induces pro-inflammatory gene expression[31], and in cancer cells, Mi-2β promotes tumor development and metastasis[63,64].

A successful anti-tumor immune response following PD-1/PD-L1 blockade is believed to require reactivation and proliferation of clone of antigen-experienced T cells in the tumor microenvironment[8,25]. Inadequate anti-tumor T-cell effector function may preclude proper T-cell function to limit the efficacy of immune checkpoint inhibitors[8,9]. Those important factors include high levels of immune suppressive cytokines or chemokines, and recruitment of immune suppressive cells, such as myeloid-derived suppressor cells (MDSCs) and regulatory T cells (Tregs)[25]. Our data show that depletion for tumor-intrinsic epigenetic modulator Mi-2β led to the changes of tumor microenvironment to fuel CD8 T cells anti-tumor immunity (Fig. 2c).

Furthermore, Mi-2β was involved in suppression of gene sets related to IFN-γ, and IFN-α signals. Loss of functional Mi-2β significantly upregulated expression of those downstream genes of IFN signaling (Fig. 3a, b).

Several chromatin regulatory factors and enzymes have been reported to be involved in regulating the transcription of interferon-stimulated genes, including EZH2[17], ARID1A[18] and KDM5B[19]. Specifically, ARID1A interacts with EZH2 and antagonized EZH2-mediated IFN responsiveness in cancer cells[18]. Mi-2β functions in chromatin assembly, genomic stability and transcriptional repression[27]. Our biochemical studies indicate that Mi-2β bound to the CXC domain of EZH2 to promote the K510 methylation of EZH2. Methylated-EZH2 subsequently activated the trimethylation of H3K27 to inhibit the transcription of ISGs in melanoma (Fig. 4g–l). Interferon-stimulated genes are crucial in inducing and recruiting effector T cells with CXCR3 chemokine receptor into tumor microenvironment to induce anti-tumor immunity[34,35,65].

Targeted therapies have significantly improved clinical outcomes in patients with various cancer such as BRAF and MEK/ERK inhibitors in metastatic melanoma[10,66,67]. Targeted therapies have been tested widely in combination with anti-PD-1 therapies, and substantially contribute to anti-tumor immunity with immunotherapy[8,68,69], including increase of tumor antigen expression[70,71], enhancement of the function of effector T cells[72,73], overcoming the immune suppressive microenvironment of tumor[74,75]. A variety of clinical trials on combination of MAPK pathway-targeted therapy and immunotherapy in advanced metastatic melanoma have been performed and evaluated[76]. However, unexpected toxicity side effects are reported in combinational clinical trials[77–79]. In addition, a study found that immune microenvironment can act as a source of resistance to MAPK pathway-targeted therapy, which is reinforced during combinational treatment; on the other hand, the increased TNF-α and tumor-associated macrophages by MAPK blockade may involve in developing immunosuppressive tumor microenvironment[80]. Here, we developed a specific and effective inhibitor of Z36-MP5 to target Mi-2β ATPase activity (Fig. 5b). With both syngeneic and transgenic mouse models, Z36-MP5 induced responses of resistant melanoma to immunotherapy of anti-PD-1 through rescue of ISG expression (Fig. 6g). Z36-MP5 is our first generation of Mi-2β-targeted inhibitor. Although Z36-MP5 is not entirely as effective as complete genetic ablation of Mi-2β, we will further improve its specificity and sensitivity by the structure modifications in the near future. The ability to target Mi-2β and recover ISG expression and inflammatory signals by Z36-MP5 developed here might be further evaluated for integration of combinational immunotherapy in patients with melanoma and other immune resistant cancers in future translational and clinical researches.

## Methods
### Plasmids and shRNAs
The plasmid of Flag-Mi-2β was generously provided by Dr. Joel Mackay in University of Sydney; the plasmid of HA-Mi-2β was synthesized by

our laboratory. To knockdown Mi-2β in B16F10 melanoma cells, mouse specific short hairpin RNAs of TRC Lentiviral Mouse Mi-2β shRNA (TRCN0000086143: TTTACAACTCAGAAGATGGGC and TRCN0000086146: TAAGTTGTGGAACCTCTCAGG) (Open Biosystems- Horizon Discovery) targeting Mi-2β were co-transfected with psPAX2 (Addgene, 12260) and pMD2.G (Addgene, 12259) into HEK293FT cells using Lipofectamine 3000. Lentiviruses were harvested 48 h after the transfection, and then used to infected B16F10 cells for 24 h in the presence of 8 μg/mL polybrene. The infected cells were selected by 2 μg/mL puromycin.

To knockdown EZH2 in melanoma cells, specific short small interfering RNA (siRNA) targeting EZH2 were transfected into B16F10 cells using Lipofectamine 3000. Protein or mRNA were harvested 48 h after the transfection. Sense of si-EZH2 are shown in Supplementary data 3.

LentiCRISPR v2 constructs for knockout mouse Mi-2β were generated following the online guide of CHOPCHOP[81] (Senses have shown in Supplementary data 3). Briefly, HEK293FT cells in 6-well plates were transfected with 1.5 μg lentiviral plasmid, 1 μg psPAX2, and 0.5 μg pMD2.G. Lentivirus were collected after 2 days after transfections, and then filtered through a 0.45 μm filter. B61F10 Cells were infected with lentivirus for 24 h, and then refed with fresh medium and selected with 2 μg/mL puromycin.

## Cell culture

B16F10 cells and primary mouse melanoma cells (extracted and saved by our laboratory) cell were cultured in complete DMEM media (10% FBS and 100U/ml of penicillin–streptomycin). B16F10-shMi-2β and B16F10-shScramble cells were maintained in complete DMEM media (10% FBS and 100 U/ml of Penicillin-Streptomycin) with 2–5 μg/ml of puromycin. CD8 T cells isolated from mice were cultured in complete RPMI 1640 media (10% FBS, 0.05 mM 2-mercaptoethanol, 20 mM HEPES, 2 mM L-glutamine, 1 mM sodium pyruvate and 100U/ml streptomycin and penicillin).

## Quantitative real-time PCR (RT-qPCR)

The total RNA was extracted with QIAGEN RNeasy kit (Invitrogen) for cDNA synthesis with SuperScript II Reverse Transcriptase (Invitrogen). In total, 40 ng cDNA was used for quantitative real-time PCR amplification by TaqMan Gene Expression Master Mix (Thermo Fisher Scientific). The relative transcript levels were normalized with GAPDH expression. The data were calculated with the comparative CT method. Primers for RT-qPCR assays are shown in Supplementary Data 3.

## Cell viability assay

B16F10 cells with shMi-2β or shScrambles were transfected with GFP expression vector pcDNA3-EGFP (Plasmid #13031), and a stable cell line was established after selection with 800 μg/mL G418 for 2 weeks. The cell viability was detected by CCK-8. B16F10 cells (shMi-2β or shScramble) were seeded into 96-well plates at a density of $1 \times 10^3$ cells/100 μl per well. The CCK-8 assay (AbMole, Beijing, China) was performed according to the manufacturer's instructions. The optical density values were measured at a wavelength of 450 nm by spectrophotometer.

To test whether Z36-MP5 affects cell proliferation, cells were seeded into 96-well plates at a density of $1 \times 10^3$ cells/100 μl per well. CCK-8 was used to detect changes in proliferation ability after 3 days of treatment with Z36-MP5 (25 μM). To test whether Mi-2β expression in T cells or treatment of T cells with Z36-MP5 affects the ability of T cells to kill tumor cells, Mi-2β in Pmel-1 T cells was manipulated or Pmel-1 cells were exposed to Z36-MP5 (25 μM) for 24 hrs. The number of cells was quantified. One thousand Pmel-1 T cells were then co cultured with 1000 tumor cells for 24 hrs, and the number of tumor cells was counted.

To test whether Mi-2β deficiency affects the sensitivity of tumor cells to Braf inhibitors, melanoma cells were isolated from *Tyr::CreER;BRaf^CA;Pten^lox/lox* and *Tyr::CreER;BRaf^CA;Pten^lox/lox; Mi-2β^lox/lox* mice and a cell proliferation assay was performed. BRAF mutation and Mi-2β-depleted melanoma cells or BRAF mutation melanoma cells ($5 \times 10^3$ cells/100 μl) were cultured with 10 μM PLX4032 (Selleck, China) for 24 h and cell proliferation of melanoma cells were determined using the CCK-8 assay.

## Immunoblot analysis

The lysis buffer (50 mM Tris pH 7.4, 1% Triton X-100, 0.5 mM EDTA, 0.5 mM EGTA, 150 mM NaCl, 10% glycerol, 1 mM phenylmethylsulfonyl fluoride and complete protease inhibitor cocktail (Roche)) were used to prepare the whole-cell lysates, which was followed by homogenization and centrifuge (14,000 rpm for 15 min at 4 °C). Pierce BCA Protein Assay Kit (Thermo Fisher Scientific) was used to detect protein concentration. After SDS-PAGE separation and PVDF membrane (BIO-RAD) transfer of the proteins, the specific primary was probed at 4 °C for overnight, before incubated with corresponding horseradish peroxidase (HRP)-conjugated 2nd antibodies. Pierce ECL Western Blotting Substrate (Thermo Fisher Scientific) was used for protein detection. Antibodies were: anti-Mi-2β(1:1000), anti-EZH2 (ab70469, Abcam) (1:1000), anti-β-actin-peroxidase antibody (AC15) (1:5000) and anti-rabbit secondary antibody (A-4914) (1:10,000). Antibody information have shown in Supplementary data 3.

## Microarray assay

Total RNA was extracted from B16F10 with Mi-2β knockout and the control cells treated with IFN-γ (10 ng/mL) for 24 hours with the RNeasy Mini Kit (74104) (Qiagen, Hilden, Germany). The experimental group cells were cultured in triplicate. The experiment was comprised of 6 Mouse Gene 2.0 ST arrays. The arrays were normalized together using the Robust Multiarray Average algorithm and a CDF (Chip Definition File) that maps the probes on the array to unique Entrez Gene identifiers. The expression values are log2-transformed by default. The technical quality of the arrays was assessed by two quality metrics: Relative Log Expression (RLE) and Normalized Unscaled Standard Error (NUSE). For each sample, median RLE values > 0.1 or NUSE values > 1.05 are considered out of the usual limit. All arrays had median RLE and NUSE values well within these limits. Benjamini-Hochberg FDR correction was applied to obtain FDR-corrected *p* values (*q* values), which represent the probability that a given result is a false positive based on the distribution of all p values on the array. In addition, the FDR q value was also recomputed after removing genes that were not expressed above the array-wise median value of at least 3 arrays (i.e., the size of each experimental group).

## ELISA assay

B16F10 cells ($1 \times 10^6$) with or without Mi-2β knockdown were seeded in 6-well plates in complete growth medium. Cell medium was changed to serum-free medium, before treatment with IFN-γ at indicated concentration for 24 h. The secreted chemokines were measured by mouse Cxcl9 ELISA kit (ab203364) and mouse Cxcl10 ELISA Kit (ab214563), according to the manufacturer's protocols. Isolated graft tumors were prepared and minced with blades, then tumor tissues were cultured in PBS (250 mg/500 μl) for 4 h at 37 °C. The secreted amount of the chemokines in the culture were measured by mouse Cxcl9 ELISA kit (ab203364) and mouse Cxcl10 ELISA Kit (ab214563), according to the manufacturer's protocols.

## Validation of genes of the epigenetic factors

The gRNA sequences targeting the selected 10 epigenetic factors (3 gRNAs/gene) were cloned into a LentiCRISPRv2GFP vector (Addgene, #82416) following the CHOPCHOP. Briefly, HEK293FT cells in 6-well plates were transfected with 1.5 μg lentiviral plasmid, 1 μg psPAX2, and

0.5 µg pMD2.G with Lipofectamine™ 3000 Transfection Reagent (ThermoFisher, #L3000001). Lentivirus were collected after 2 days of transfections. After filtered through a 0.45 µm filter, the lentivirus was stored at −80 °C. B61F10 cells were infected with lentivirus for 24 hours individually. Infected cells were sorted based on GFP expression by BD FACS Aria II. Cells (1.5×105) were mixed with BD matrigel (Matrix Growth Factor Reduced) (BD, 354230) in 100 µl PBS, and then subcutaneously injected into the right flanks of C57BL/6 mice of 8–10 week old (from the Jackson Laboratory, 000664). Tumor weight was counted and were extract tumor-infiltrating T cells after 18 days.

## Assay for transposase-accessible chromatin with high-throughput sequencing (ATAC-seq) and data analysis

ATAC-seq was conducted in Frasergen (Wuhan, China) as previously reported[82] with minor modifications. Briefly, $5 \times 10^4$ cells (B16F10-shMi-2β and B16F10-shScramble cells) were resuspended in ice-cold nucleus lysis buffer (10 mM Tris pH 7.4, 10 mM NaCl, 3 mM MgCl$_2$, and 0.1% IGEPAL CA-630) and centrifuged at $500 \times g$ for 10 min at 4 °C. Centrifuge $500 \times g$ at 4 °C for 10 min, carefully remove the supernatant; Cell precipitation was cleaned with pre-cooled buffer solution, centrifuged at $500 \times g$ at 4 °C for 10 min, and the nucleus was collected. Then, appropriate amount of nucleus was taken, Tn5 transposase was added, and the reaction was carried out at 37 °C for 30 min. DNA fragments were recovered and amplified by PCR. Agilent2100 was used to detect the size distribution of library fragments. For the completed library, 1 µl was taken for Agilent2100 test. According to the Agilent2100 library check, the quality control results of the sequencing data were judged to be qualified, and the next data analysis was carried out. After qualified library inspection, different libraries were mixed according to the requirements of effective concentration and target data volume, and Illumina PE150 sequencing was performed. The original image Data files obtained by Illumina sequencing platform are converted by Base Calling analysis into Sequenced Reads, which we call Raw Data or Raw Reads. The results were stored in FASTQ (fq) file format, which contained sequence information of sequencing sequences (Reads) and their corresponding sequencing quality information. Reads were mapped to the mm10 reference genome. Peaks with log2 (fold change) ≥ 0.5 and a $P$ value ≤ 0.05 in comparisons were termed significant. Genome coverage (bedgraph) files were generated by the makeTagdirectory with checkGC parameter, and were used for visualization with IGV2. Read distribution (RD) plots were visualized by the Java TreeView, and histograms were visualized by the GraphPad Prism 5 software (GraphPad Software Inc., San Diego, CA). The ATAC-seq dataset have been deposited in the GEO under the GEO Series ID is GSE255782.

## Syngeneic melanoma graft mouse model

Mi-2β knockdown or Scramble B16F10 cells ($1.5 \times 10^5$) were mixed with BD matrigel (Matrix Growth Factor Reduced) (BD, 354230) in 100 µl PBS, and then subcutaneously injected into the right flanks of C57BL/6 mice of 8–10 week old (from the Jackson Laboratory, 000664). Tumor growth was measured with calipers, and size was expressed in cubic centimeters every 3 days. For antibody treatment, control IgG (10 mg/kg) or anti-PD-1 (RMP1-14, BioXCell, 10 mg/kg) was injected intraperitoneally (i.p.) on day 6, 9, 12, 15 and 18 after tumor cell inoculation. For tumor growth curve, grafts were measured with calipers and established (0.5*length × width$^2$) every three days. For survival tests, mice were euthanized when the tumor size exceeded 1 cm$^3$. To test Z36-MP5 function in syngeneic mouse model, B16F10 cells ($1.5 \times 10^5$) were mixed with BD matrigel (Matrix Growth Factor Reduced) (BD, 354230) in 100 µl PBS, and then mouse subcutaneous injection and tumor graft monitor were performed as described above. Except that vehicle [5% (w/v) Kolliphor HS 15 (Sigma)] in normal saline or formulated 30 mg/kg Z36-MP5 or 150 mg/kg GSK126 (aladdia) was administered with i.p.

injection once a day starting at day 6, together with i.p. injection of control IgG (10 mg/kg) or anti-PD-1 (RMP1-14, BioXCell, 10 mg/kg) on day 6, 9, 12, 15 and 18. The mice were euthanized after indicated days or when the allowable endpoint size (1 cm$^3$) was reached. All mice were maintained in pathogen-free conditions in the animal facility at Boston University. All animal experiments were performed in accordance with the Guide for the Care and Use of Laboratory Animals of the National Institutes of Health, and the protocol was reviewed and approved by the Animal Science Center (ASC) of Boston University.

## Antibody-mediated depletion and Z36-MP5 treatment

To test Z36-MP5 function in antibody-mediated CD8+ T cells depletion mouse model, B16F10 cells ($1.5 \times 10^5$) were mixed with BD matrigel in 100 µl PBS, and then mouse subcutaneous injection and tumor graft monitor were performed as described above. Except that formulated 30 mg/kg Z36-MP5 was administered with i.p. injection once a day starting at day 6, together with i.p. injection of control IgG antibodies, 200 µg anti-CD4 (BioXCell, BE0003-1, clone GK1.5) or 200 µg anti-CD8 (BioXCell, BE0061, clone 2.43) antibodies on day 3, 6, 9, 12, 15 and 18. The mice were euthanized after indicated days or when the allowable endpoint size (1 cm$^3$) was reached.

## Genetically engineered mouse models

*Mi-2β$^{lox/lox}$* mice were generated and generously provide by Dr. Georgopoulos lab (Massachusetts General Hospital at Harvard Medical School)[31]. Tyr::CreER;*BRaf$^{CA}$;Pten$^{lox/lox}$* mice were purchased from Jackson laboratories (Stock No: 013590). All strains of mice were on the background of C57BL/6J background. Gene activation and silencing were induced with intraperitoneal (i.p.) administration of 100 µL/mouse/day tamoxifen (20 mg/mL) for constant 5 days. Mice with measureable tumors were randomly treated with either control IgG antibodies (10 mg/kg) or anti-PD-1 (RMP1-14, BioXCell, 10 mg/kg) by i.p. administration at day 9, 12, 15, 18 and 21 after Cre activation. To test Z36-MP5 function in vivo, vehicle [5% (w/v) Kolliphor HS 15 (Sigma)] in normal saline or formulated 30 mg/kg Z36-MP5 was administered with i.p. injection once a day starting at day 9 after Cre activation, together with i.p. injection of control IgG antibodies (10 mg/kg) or anti-PD-1 (RMP1-14, BioXCell, 10 mg/kg) starting on day 9, 12, 15, 18 and 21 after Cre activation, as indicated. Tumor growth was then monitored each the other day. All mice were bred and maintained in pathogen-free conditions in the animal facility at Boston University. When the following conditions occur, the mouse is regarded as dead and euthanized, including weight loss greater than 15%; significant abdominal distension, especially if it begins to compromise respiratory ability of animal; hunched posture; failure to eat or drink; absence of (or abnormal) feces or urine output; reluctance to move or abnormal gait; Discharges or hemorrhage. Abnormal behavior or vocalizations and tumor size exceeded 1 cm$^3$. All animal experiments were done according to protocols approved by the ethics committee review board of Boston University and in accordance with the guidelines set forth by the US National Institutes of Health and The ethics committee review board of The 2nd Hospital and School of Medicine, Zhejiang University. We statement confirming that the maximal tumor size/burden was not exceeded of the maximal tumor size/burden permitted by ethics committee review board.

## Kaplan−Meier survival analysis

TCGA dataset was downloaded from website (http://tcgabrowser.ethz.ch:3839/TEST/). The melanoma patients (n = 454) were divided into CD8 High and CD8 Low groups based on the mRNA expression of CD8. The median gene expression of CD8 was set as the cutoff. For each Gene and CD8 High/Low group, we further divide the samples into High and Low subgroups based on the gene's median expression. The Kaplan-Meier survival curves were generated, and their differences were examined using a log-rank test.

## Chromatin immunoprecipitation (ChIP) assays

Briefly, B16F10 cells (~1 × 10$^7$) were incubated with 1% formaldehyde for 10 min for crosslink, with adding glycine for a final concentration of 0.125 M to stop crosslink. Then the nuclear pellets were prepared, and suspended with ChIP lysis buffer. The DNA was fragmented with sonication. Immunoprecipitation was performed with antibodies anti-Mi-2β (ab70469, Abcam), anti-Stat1 (ab239360, Abcam) and IgG control at 4 °C for overnight. The complex was pulled down with A/G agarose beads (#20422, Thermo Fisher Scientific) and crosslink was reversed with heating at 65 °C for overnight. The DNA was purified and eluted for quantitative PCR assay. Primers were designed based on the binding peak analysis with ChIP-Atlas-Peak Browser. All data were normalized to gene desert regions of the IgH loci. The real-time PCR was performed in triplicate. Values of [Δ][Δ] Ct method was used to calculate the relative binding enrichment, with the formula: Ct, template (antibody) − Ct, template (IgG) = [Δ] Ct, and the fold enrichments ([Δ][Δ]Ct) were determined using the formula of 2-[Δ] Ct. (experimental)/2-[Δ] Ct (IgH). Standard error from the mean was calculated from replicate [Δ][Δ] Ct values from independent experiments. Primers for ChIP assays are shown in Supplementary data 3.

## Preparation of tumor-infiltrating T cells

Tumors were minced with scissors, and then digested with the digestion buffer (RPMI 1640 medium, 5% FBS, 1% penicillin-streptomycin, 25 mM HEPES, and 300 U collagenase (Sigma C0130)) on a shaker at 37 °C for 2 hours. Single cells were prepared through a 70 μm cell strainer. Erythrocytes were removed by incubation in red blood cell lysis buffer (R7757, Sigma) at room temperature for 5 min. The cells were prepared in PBS (with concentration of ~2 × 10$^7$) for studies.

## Flow cytometry

The single-cell suspension were fixed with 2% paraformaldehyde solution (J19943K2, Thermo Scientific). CD45 + T cells were selected from single-cell suspension (about 2 × 10$^7$ cells digest from 002 g tumor tissue) by flow cytometry with the follow antibodies (anti-mouse CD45; BD pharmingen, 561875). And then the cells were stained anti-mouse CD3e PE (145-2C11, BD pharmingen, 553063), anti-mouse CD4 FITC (RM4-5, BD pharmingen, 553046), anti-mouse CD4 PE/Cy7 (GK1.5, BioLegend, 100421), anti-mouse CD8 FITC (53-6.7, BD pharmingen, 553031), anti-mouse CD8a APC/Cy7 (53-6.7, BioLegend 100713), anti-mouse IFN-γ PE (XMG1.2, eBioscience, 12731181), anti-mouse CD69 PE (H1.2F3, Biolegend, 104508), anti-mouse CD25 Alexa Fluor 488 (PC61.5, eBioscience, 53025182), anti-mouse CD107a-V450 (1D4B, BD, 560648), anti-human/mouse granzyme B FITC (GB11, BioLegend, 515403). The regulatory T cells in TILs were stained with the Mouse Regulatory T Cell Staining kit#1 (88-8111, ThermoFisher Scientific), with antibodies of anti-mouse CD4 FITC (RM4-5), anti-mouse CD25 APC (PC61.5) anti-mouse Foxp3 PE (FJK-16s). BD LSRII was used for data acquisition and FlowJo was used for data analysis.

## Protein expression and purification

Flag-Mi-2β was expressed and purified from HEK293 cells, which were cultured in DMEM supplemented with 10% Fetal Bovine Serum 100 unites/ml penicillin and 100 μg/ml streptomycin. Flag-Mi-2β in pcDNA3.1 expression vector were transfected into HEK293 cells with Lipofectamine™ 3000 Transfection Reagent (ThermoFisher) for 3 days. The resulted cells were harvested for the nuclear pellet extraction with cytoplasmic lysis buffer (50 mM HEPES, 10 mM KCl, 1.5 mM MgCl$_2$, 1 mM DTT, 1 mM PMSF and 1× protease inhibitor, pH 7.5) on ice for 30 minutes. The nuclear pellet was collected by spun down. The nuclear lysis buffer (50 mM HEPES, 0.5 M NaCl, 1 mM EDTA, 1% Triton X-100, 1.5 mM MgCl$_2$, 1 mM DTT, 1 mM PMSF, and 1× protease inhibitor, pH 8) was used to resuspend nuclear pellet for homogenization by sonication. Nuclear extract was incubated with Flag-Mi-2β affinity gel beads (Sigma-Aldrich) at 4 °C for overnight. The Flag M2

beads were washed, and Flag-Mi-2β protein was eluted with 300 μg/ml 3XFlag peptide (Sigma-Aldrich), in 20 mM HEPES, 150 mM NaCl, 1 mM DTT, and 10% glycerol, pH 7.5. Protein was confirmed by SDS-PAGE and coomassie stains. All the purified protein samples were concentrated, aliquoted and flash-frozen in liquid nitrogen, and then stored in −80 °C for later use.

## TCGA data analysis

To analysis the hazard ratio of epigenetic factor in human melanoma samples, we downloaded the ATGC dataset of melanoma from http://tcgabrowser.ethz.ch:3839/TEST/ on 2018-09-03. Data of 454 melanoma patient samples were available for analysis. The patients were divided into CD8A high and CD8A low groups based on the gene expression of CD8A. The median CD8A expression was chosen as the cutoff.

## ATP-driven nucleosome remodeling reactions

The function of chromatin-remodeling enzyme was studied with EpiDyne-FRET (EpiCypher, SKU: 16-4201) according to the protocol. Briefly, Nucleosomes were assembled with the recombinant nucleosome substrates Cy5-labeled human histone octamer (H2A T120C-Cy5) wrapped with 5′ Cy3-labeled DNA (207 bp), in which contains a terminally nucleosome positioning Widom 601 element. Cy3-Cy5 FRET is at a maximum level at the assembled starting state. When the histone octamer is relocated towards the DNA 3′ by chromatin remodeler enzymes, Cy3-labeled DNA 5′ end is moved away from the Cy5-labeled octamer, leading to a reduction in FRET signal. The optimal conditions of the Mi-2β enzyme and the ATP concentrations in the 96-well were determined using FRET signal which was read by QuantStudio 12 K Flex Real-Time PCR System with capable of Cy3 (Excitation-531 nm/Emission-579 nm)/Cy5 (emission-685 nm) detection. Data is expressed as the ratio of the raw Cy3 and Cy5 emission signals at each time point. For the Mi-2β concentration and reaction time optimization, Flag-tagged Mi-2β at series of concentrations (ranging from 0.4 to 250 nM), ATP at a non-limiting concentration (1 mM) were added to 96-well white solid plates and incubated for different times (0 to 50 min) with the substrate EpiDyne-FRET nucleosome at a saturated concentration (20 nM), in the 50 μL reaction buffer containing 50 mM Tris, pH 7.5, 50 mM KCl and 3 mM MgCl$_2$. The nucleosome remodeling reaction was stopped by adding 10 mM EDTA and 0.25 mg/ml Salmon Sperm DNA. The assay had a sufficiently high assay signal, and a minimal substrate conversion for a sufficient assay window was taken. We finally chose 12.5 nM Mi-2β and a reaction time of 15 minutes as the optimal condition for the nucleosome remodeling assay. The ATP titration was performed with Mi-2β using the enzyme concentration and reaction time previously determined, with at ATP concentrations ranging from 0.1 to 300 μM. The Michaelis-Menten equation was performed to calculate the apparent ATP Km. At the ATP concentration of 11.54 μM, Mi-2β showed a 50% change between the maximum and minimum reaction signal levels.

Z-factor was used to determine the assay quality (Z-factors above 0.5 represent an assay with an excellent quality). In the optimization assay procedure, the wells without Mi-2β was defined as 100% inhibition controls, and that containing Mi-2β was regarded as the 0% inhibition controls. The FRET signaling in each well was detected and Cy3/Cy5 ratio was calculated. Then the average (represented as μ) and standard deviations (represented as σ) of the ratios were calculated too. The Z-factor equation is Z-factor = 1 − 3 × ($\sigma_{0\%Inhibition}$ + $\sigma_{100\%Inhibition}$)/($\mu_{0\%Inhibition}$ − $\mu_{100\%Inhibition}$). The Z-factor was 0.729 for Mi-2β, which confirmed the optimization of assay conditions including enzyme concentration, ATP concentration and the reaction time.

## Homology modeling and virtual screening

Homology Modeling was carried out using Structure Prediction Wizard in Prime. The Homology Model of Mi-2β (CHD4) was generated

using the yeast CHD1 structure (PDB code:3MWY) as template and the receptor sequence was obtained from Uniprot. Standard options were used when running the program and one homology model was gotten. For the output structure, the receptor was properly prepared using Protein Preparation Guide. Virtual screening was done in the default workflow process. First, enzyme hinge region ligands database and nucleoside mimetics database from Enamine are prepared using a LigPrep and 3 low energy conformations are generated for each ligand. Then all ligands are docked to the ATP-binding site for Mi-2β using SP docking and postprocessed with Prime MM-GBSA. After minimization, we kept top 1000 ligands from MM-GBSA score for each database. We have identified ligands with methyldihydroimidazopyridinone structure can interact well with the ATP warhead binding region of Mi-2β.

### Profile of Z36-MP5 inhibition on ATPases
The Profile of Z36-MP5 inhibition on ATPases was measured by ActivX Biosciences inc. (La Jolla, CA). In briefly, Z36-MP5 was directly added to A375 cell lysates generated with a tip sonicator, and the resulting lysate was clarified by centrifugation at $16,100 \times g$ for 15 minutes to get the native cell lysate. For the ATP acyl phosphate probe-based chemoproteomics, lysine residues in ATP-binding sites were acylated with a desthiobiotin tag, and labeled peptides were isolated by affinity capture. The probe labeling reaction could be blocked by ATPase inhibitors. Labeled peptides were identified on the basis of their MS spectra generated by data-dependent LC-MS/MS. Duplicated treated samples and control samples were performed and the inhibition results were analyzed as % changes with statistically significance (Student $t$-test score <0.05).

### Co-immunoprecipitation and mass spectrometry
In brief, B16F10 cells ($1 \times 10^8$) were collected and washed three times with ice-cold phosphate-buffered saline (PBS) and lysed in lysis buffer (50 mM Tris pH 7.4, 1% Triton X-100, 0.5 mM EDTA, 0.5 mM EGTA, 150 mM NaCl, 10% Glycerol, 1 mM phenylmethylsulfonyl fluoride and complete protease inhibitor cocktail (Roche)) on ice for 30 min. The cell lysates were centrifuged and collected at $15,000 \times g$ for 15 min at 4 °C. Supernatant (500 µl) was incubated with Protein G Agarose Beads (Thermo Fisher Scientific) 30 min at 4 °C and collected at $500 \times g$ for 5 min at 4 °C. Supernatant incubated with different primary antibodies overnight at 4 °C. Protein G Agarose Beads were added into supernatant for incubation with rotation at 4 °C for 1 h. After three washes with 1 mL of lysis buffer, the supernatant were collected at $500 \times g$ for 5 min at 4 °C and prepared protein samples were incubated with sample buffer for 10 min at 100 °C. The bound proteins were resolved by SDS-PAGE and immunoblotted with indicated antibodies.

For identification of Mi-2β-interacting proteins, B16F10 cells ($1 \times 10^8$) were collected and washed three times with PBS and lysed in lysis buffer on ice for 30 min. Cell lysates were collected and then isolated overnight by anti-Mi-2β magnetic agarose beads. The anti-Mi-2β magnetic agarose beads were collected and washed three times with PBS. The prepared protein samples were incubated with sample buffer for 15 min at 100 °C, and then separated by SDS-PAGE (10%). The gel was immersed in staining solution (0.3% Coomassie blue, 45% methanol, 10% glacial acetic acid and 45% dH₂O) on shaker for 30 min, followed by incubation in destaining solution (20% methanol, 10% glacial acetic acid, and 70% dH2O) on the shaker overnight. The bands were excised and sent to Biological Mass Spectrometry Facility of Shanghai Applied Protein Technology Co., Ltd for protein identification.

### Pharmacokinetics of Z36-MP5 in rats
Compound Z36-MP5 was evaluated in a pharmacokinetic study in male Sprague-Dawley (SD) rats following intraperitoneal injection of Z36-MP5 at 1.0 mg/kg as a solution in 5% DMSO, 30% PEG400, and 65% corn oil. Blood was collected at 0.25, 0.5, 1, 2, 4, 8, and 24 h following intraperitoneal injection. The blood samples were placed in wet ice, and serum was collected after centrifugation. Serum samples were frozen and stored at −80 °C. The serum samples were analyzed utilizing HPLC-coupled tandem mass spectrometry. Values are calculated from arithmetic mean plasma concentrations ($n = 3$ rats per condition). Rats experiments were done according to protocols approved by the ethics committee review board of University of Arkansas for Medical Science and in accordance with the guidelines set forth by the US National Institutes of Health.

### Chemical synthesis Z36-MP5
Flash chromatography was performed using silica gel (200–300 mesh). All reactions were monitored by thin-layer chromatography (TLC) on silica gel plates. ¹H-NMR spectral data were recorded on Varian Mercury 400 NMR spectrometer, and ¹³C-NMR was recorded on Varian Mercury 126 NMR spectrometer at ambient temperature. Chemicals shifts ($\delta$) were reported in ppm, coupling constants (J) were in hertz, and the splitting patterns were described as follows: s for singlet; d for doublet; t for triplet; q for quartet; and m for multiplet. Mass spectrometry was conducted using a Thermo Fisher LCQ-DECA spectrometer (ESI-MS mode). All tested compounds were purified to ≥95% purity as determined by high performance liquid chromatography (HPLC). Reagents and conditions please see Supplementary Fig. 11.

### Statistical analysis and study design
Animals were grouped randomized. The qualification experiments were blinded by investigators. All samples or animals were included in analysis. The unpaired, two tailed t-test Comparisons were performed between two groups. Statistical tests were done with biological replicates. $P < 0.05$ was considered statistically significant. $*P < 0.05$, $**P < 0.01$, $***P < 0.001$.

### Reporting summary
Further information on research design is available in the Nature Portfolio Reporting Summary linked to this article.

## Data availability
The Microarray assay dataset and ATAC-seq dataset used in this study are available in the GEO database under accession code is GSE151640 and GSE255782. The TCGA data of human melanoma samples used in this study were obtained Tumor IMmune Estimation Resource [https://cistrome.shinyapps.io/timer/]. The Mass spectrometry data have been deposited to the ProteomeXchange Consortium via the iProX repository with the dataset identifier PXD049467 [https://www.iprox.cn//page/project.html?id=IPX0008199000]. The remaining data are available within the Article, Supplementary Information or Source Data file. Source data are provided with this paper.

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

## Acknowledgements

R.C. was supported by National Key Research and Development Program of China (Grant Nos. 2021YFA1101000, 2021YFA1101004), National Natural Science Foundation of China (Grant No. U21A20379), Research Fund for Foreign Scholars of China (Grant No. 82250710176) and Leading innovation and entrepreneurship team of Hangzhou (TD2020006). C.R.G. was funded by the Ludwig Institute for Cancer Research. Z.X. was supported by National Key Research and Development Program of China (Grant No. 2023YFE0109800).

## Author contributions

R.C., H.L. and X.Liao designed the research and R.C., H.L., X.Liao and M.X. supervised the study. C.L., Z.W., and L.Y. designed and performed the initial experiments. X. Lin, J.S., Z.J., X.L., J.R.H and P.D.T cooperated the homology modeling and virtual screening. Y.C. and M.C. cooperated syngeneic melanoma graft mouse model. Y.L; Y,J and Z.X. performed the ATAC-seq data analysis. C.L., C.R.G. and R.C. wrote the manuscript with contributions from X.M., H.L. and X.Liao.

## Competing interests

The authors declare no competing interests.
