## [Peer Review File · Nature Communications]

Mi-2 β promotes immune evasion in melanoma by activating EZH2 methylationEditorial Note: This manuscript has been previously reviewed at another journal that is not operating a transparent peer review scheme. This document only contains reviewer comments and rebuttal letters for versions considered at *Nature Communications*.

REVIEWER COMMENTS

Reviewer #1 (Remarks to the Author):

Li et al. identify the chromatin remodeling enzyme Mi-2B as a key melanoma-intrinsic chromatin regulatory factor whose expression is inversely correlated with anti-tumor T cell responses. Using a subcutaneous as well as a genetically engineered cancer model, they demonstrate that loss of Mi-2B sensitizes melanoma cancer cells to anti-PD-1 therapy, resulting in an effective anti-tumor immune response. The original manuscript identified a role for Mi-2B in the regulation of the IFN γ -responsive genes and that loss of Mi-2B enhances the expression of, amongst others, the T cell chemoattractants CXCL9 and CXCL10, leading to increased CD8 T cell infiltration into tumors. In the revised manuscript the role of Mi-2B is explored in more detail by using ATAC-sequencing, ChIP-sequencing, and mass spectrometry experiments. Now the authors show that Mi-2B interacts with EZH2, promoting EZH2 auto-methylation, which is required to enhance its H3K27 trimethylation activity. Furthermore, Li and colleagues develop an inhibitor of Mi-2B, Z36-MP5, which induces a response to anti-PD-1 therapy in otherwise resistant melanomas. Taken together, this is a well-performed study that demonstrates a role for an epigenetic regulator in the regulation of the immunotherapy response against melanoma.

Minor comments/suggestions:

1. The authors convincingly demonstrate an interaction between Mi-2B and EZH2 and that Mi-2B promotes the activity of EZH2, leading to increased global H3K27me3 levels (Supplemental Figure 4d-e). Furthermore, they demonstrate that depletion of Mi-2B increases the chromatin accessibility of the Cxcl9 and Cxcl10 promoter. This suggests that chromatin accessibility is increased in shMi-2B cells due to reduced H3K27me3 in these promoter regions, but this is not experimentally addressed. Can the authors look at H3K27me3 levels specifically in the Cxcl9 and Cxcl10 promoter region in WT and shMi-2B

cells (for example by CHIP-qPCR)?

2. Fig 2b: The text states: "There was no significant difference in mouse survival observed on a BRafV600E/Ptenu null background irrespective of Mi-2 β status (Fig. 2b)." However, Fig 2b demonstrates a significant difference between BRafV600E/Ptenu null and BRafV600E/Ptenu nullMi-2 β null mice.

3. Supplemental Figs 2a and 4c: Quantification of IHC data would be ideal here (as done for Supplemental Fig 2e CD8 T cells).

4. Supplemental Fig 6j: It does not appear to me that expression of the Ezh2 K to A mutants actually exhibit reduced H3K27me3 as the text states. Can this be quantified?

5. Fig 4i-j: I found the stated impact in text of the K510A mutation in Ezh2 impacting its own monomethylation hard to follow from the data presented here. While it is clear to me that K735A does not impact Kme1, Kme1 looks prevalent in the K510A mutant experiments. Notably, the Kme1 is high in the negative control lanes too, which I suppose is why the lack of increase in Kme1 is interpreted to mean that K510-methylation is critical here. A clear description of the interpretation of the results would be helpful here.

6. TIMER analysis demonstrates a negative correlation between Mi-2B mRNA levels and CCL5, CD74 and CD40 mRNA levels in melanoma patients. What about CXCL9 and CXCL10 levels? Why aren't these reported?

7. In the text, "GSK126" is mis-spelled "SGK-126" although spelled correctly in Supplemental Fig 10.

Reviewer #2 (Remarks to the Author):

Authors provided more details on the manuscript, particularly with more mechanistic insights. However, the characterization of the compound on target effect is still not enough to convince the effect is on target. Especially with the ATP competitive data in figure 5e. with the normally cellular ATP concentration been around 1-10 mM, the compound will be have very low inhibitory activity, and the inhibitory IC50 to the mi2-b would be 10 uM or even higher, which are not supporting the on-target effect observed with compound treatment at 5 uM or even 25 uM. When the selectivity profiling performed in supplement table 3, mi-2b (or CHD4) is not included for some reason. it should not be compared with

the IC50 from other binding assay to show as selectivity.

Reviewer #3 (Remarks to the Author):

Overall, the authors have significantly revised their manuscript and answered most of the reviewer comments. But the clarity with regards to the mechanism behind immune sensitization of Mi-2B loss is still not sufficient and there are still three major points to be addressed before this paper is acceptable for publication:

1. While they show how Mi-2B loss activates the ISGs via EZH2 repression, they don't establish that this is the mechanism for the anti-tumor/ CD8 activation response. Akin to the figure S8f, they need to demonstrate that cells lacking EZH2 do not show a combinatorial effect when treated with anti-PD1 and Mi-2B inhibition.
2. All their flow data is still in percentage of CD4/CD8s and since the tumor sizes of the combination treatment in particular is much smaller, this is not an accurate representation. They need to show counts of these cells normalized by tumor weight, their counts from IHC alone is not sufficient to answer this critique.
3. The results with Z36-MP5 differ from their results using Mi-2B loss (shRNA or genetic loss)- the combinatorial effect with anti-PD1 is lower probably because there is no effect on CD4/T regs. The latter effect is consistently seen with both forms of genetic ablation of Mi-2B. They need to provide an explanation for this difference. For example, by comparing degree of EZH2 inhibition or activation of ISGs between the shRNA/genetic ablation of Mi-2B and the drug.

Minor points:

- They refer to B16 implantation in B6 mice incorrectly as a xenograft model in a few places
- Line 335 is confusing as it suggests that Mi-2B loss promotes T cell mediated cytotoxicity in vitro but their data doesn't support it and they reiterate in the rebuttals that Mi-2B loss doesn't alter efficacy of T cell killing.

"Together, all these data suggest that Z36-MP5 is a potent and effective inhibitor 335 for Mi-2 β and stimulates T cell mediated cytotoxicity in vitro, which warranted further in vivo studies."

- Line 378 is only true with respect to B16 cells. In PM cells, even Z36-MP5 does increase H3K27ac levels, although the degree of increase may be less than that of GSK126. This requires a quantification and rewording on the statement to reflect the actual data.

"We confirmed these results, and found that Z36-MP5 repressed the level of H3K27me3, but did not activate H3K27ac"

REVIEWER COMMENTS

Reviewer #1 (Remarks to the Author):

Li et al. identify the chromatin remodeling enzyme Mi-2B as a key melanoma-intrinsic chromatin regulatory factor whose expression is inversely correlated with anti-tumor T cell responses. Using a subcutaneous as well as a genetically engineered cancer model, they demonstrate that loss of Mi-2B sensitizes melanoma cancer cells to anti-PD-1 therapy, resulting in an effective anti-tumor immune response. The original manuscript identified a role for Mi-2B in the regulation of the IFN γ -responsive genes and that loss of Mi-2B enhances the expression of, amongst others, the T cell chemoattractants CXCL9 and CXCL10, leading to increased CD8 T cell infiltration into tumors. In the revised manuscript the role of Mi-2B is explored in more detail by using ATAC-sequencing, ChIP-sequencing, and mass spectrometry experiments. Now the authors show that Mi-2B interacts with EZH2, promoting EZH2 auto-methylation, which is required to enhance its H3K27 trimethylation activity. Furthermore, Li and colleagues develop an inhibitor of Mi-2B, Z36-MP5, which induces a response to anti-PD-1 therapy in otherwise resistant melanomas. Taken together, this is a well-performed study that demonstrates a role for an epigenetic regulator in the regulation of the immunotherapy response against melanoma.

Response: We thank the reviewer for recognizing a well-performed study of our work.

Minor comments/suggestions:

1. The authors convincingly demonstrate an interaction between Mi-2B and EZH2 and that Mi-2B promotes the activity of EZH2, leading to increased global H3K27me3 levels (Supplemental Figure 4d-e). Furthermore, they demonstrate that depletion of Mi-2B increases the chromatin accessibility of the *Cxcl9* and *Cxcl10* promoter. This suggests that chromatin accessibility is increased in shMi-2B cells due to reduced H3K27me3 in these promoter regions, but this is not experimentally addressed. Can the authors look at H3K27me3 levels specifically in the *Cxcl9* and *Cxcl10* promoter region in WT and shMi-2B cells (for example by ChIP-qPCR)?

Response: We thank the reviewer for the constructive suggestion. We fully agree with the reviewer that it is necessary to identify the H3K27me3 levels specifically in the *Cxcl9* and *Cxcl10* promoter region in WT and shMi-2 β cells. To this end, we performed a ChIP-qPCR to identify the H3K27me3 levels specifically in the *Cxcl9* and *Cxcl10* promoter region in WT and shMi-2 β cells. We found the H3K27me3 levels at the *Cxcl9* and *Cxcl10* promoter region was indeed repressed in cells with Mi-2 β silencing (please see Fig 4e in the resubmitted manuscript). This result further confirmed that the chromatin accessibility of the *Cxcl9* and *Cxcl10* promoter was activated in cells with Mi-2 β silencing.

2. Fig 2b: The text states: “There was no significant difference in mouse survival observed on a BRafV600E/Ptenuil background irrespective of Mi-2 β status (Fig. 2b).”

However, Fig 2b demonstrates a significant difference between BRafV600E/Pten^{null} and BRafV600E/Pten^{null}Mi-2 β ^{null} mice.

Response: We thank the reviewer for pointing out this typo error and apologize for this. The correct text should be: There was no significant difference in mouse survival observed on a BRaf^{V600E}/Pten^{null} background irrespective of anti-PD-1 treatment (Fig. 2b). This typo error has been corrected.

3. Supplemental Figs 2a and 4c: Quantification of IHC data would be ideal here (as done for Supplemental Fig 2e CD8 T cells).

Response: We thank the reviewer for the suggestion. We have added the quantification of IHC data. (please see Fig S4c in the resubmitted manuscript)

4. Supplemental Fig 6j: It does not appear to me that expression of the Ezh2 K to A mutants actually exhibit reduced H3K27me3 as the text states. Can this be quantified?

Response: We thank the reviewer for pointing out this error. It's a label error. We apologize for this. It has been fixed in the resubmitted manuscript.

5. Fig 4i-j: I found the stated impact in text of the K510A mutation in Ezh2 impacting its own monomethylation hard to follow from the data presented here. While it is clear to me that K735A does not impact Kme1, Kme1 looks prevalent in the K510A mutant experiments. Notably, the Kme1 is high in the negative control lanes too, which I suppose is why the lack of increase in Kme1 is interpreted to mean that K510-methylation is critical here. A clear description of the interpretation of the results would be helpful here.

Response: We thank the reviewer for the comments. We have repeated the experiments presented in Fig 4i-j. Specifically, in the repeated experiments, we used the same exposure time in our WB experiment and added a control group. As demonstrated in Fig. 4i-j, we found that the K510 EZH2 methylation was Mi-2 β dependent. This result is consistent with our original results. In our previous result, the WB exposure time was different, which cannot compare the expression in different panels of WB experiment.

6. TIMER analysis demonstrates a negative correlation between Mi-2B mRNA levels and CCL5, CD74 and CD40 mRNA levels in melanoma patients. What about CXCL9 and CXCL10 levels? Why aren't these reported?

Response: We thank the reviewer for the helpful suggestion. TIMER analysis indicated that there was a negative correlation tendency between Mi-2 β and Cxcl9 and Cxcl10 at the mRNA level in melanomas in TCGA. However, the correlation did not reach the statistical significance. Please see the analytic results, below. These results were mentioned in the revised manuscript (lines 210-211).

7. In the text, “GSK126” is mis-spelled “SGK-126” although spelled correctly in Supplemental Fig 10.

Response: We thank the reviewer for pointing out this typo. It has been fixed.

Reviewer #2 (Remarks to the Author):

Authors provided more details on the manuscript, particularly with more mechanistic insights. However, the characterization of the compound on target effect is still not enough to convince the effect is on target. Especially with the ATP competitive data in figure 5e. with the normally cellular ATP concentration been around 1-10 mM, the compound will be have very low inhibitory activity, and the inhibitory IC50 to the mi2-b would be 10 uM or even higher, which are not supporting the on-target effect observed with compound treatment at 5 uM or even 25 uM. When the selectivity profiling performed in supplement table 3, mi-2b (or CHD4) is not included for some reason. it should not be compared with the IC50 from other binding assay to show as selectivity.

Response: We thank the reviewer for raising this concern. As the ATP-competitive biochemical assay may fail to fully replicate the physiological complexity of the full-length kinase protein in the presence of the cellular milieu and regulatory circuits that modulate kinase function and cellular microenvironment, it may not be sufficient to estimate the cellular on-target inhibitory activity of a compound merely basing on the readout of biochemical assay. There are multiple examples of compounds which showed unexpected higher on-target cellular activity than the readouts from competitive biochemical assay. For instance, structure-guided drug designs for FLT3-WT or FLT3-ITD mutant kinase inhibitors have reported significantly higher cellular potency than that obtained for kinase biochemical assay with ATP concentrations lower than 10 μ M (Tomáš Gucký et al., Discovery of N2-(4-amino-cyclohexyl)-9-cyclopentyl-N6-(4-morpholin-4-ylmethyl-phenyl)-9H-purine-2,6-diamine as a potent

FLT3 kinase inhibitor for acute myeloid leukemia with FLT3 mutations. *J Med Chem*, 61, 3855-3869 (2018); Lexian Tong et al., Identification of 2-aminopyrimidine derivatives as FLT3 kinase inhibitors with high selectivity over c-KIT. *J Med Chem*, 65, 3229-3248 (2022)). In these cases, the cellular form of the FLT3-WT or FLT3-ITD mutant was preferentially bound. Another example is paclitaxel, a microtubule-stabilizing agent that is widely used in cancer chemotherapy. Biochemical competitive binding assay suggests that paclitaxel bind with GMPcPP-stabilized microtubules with binding affinity at micromole concentrations, whereas paclitaxel shows drastically higher cytotoxicity against PC3 prostate cancer cells with IC50 value 1.41 nM (Shubhada Sharma, et al., Dissecting paclitaxel–microtubule association: quantitative assessment of the 2'-OH group. *Biochemistry*, 52, 2328-2336 (2013)). Paclitaxel sensitizes BG-1 ovarian cancer cells to radiation at a dose that was not cytotoxic and did not cause cell cycle perturbations (Albert Steren, et al., Taxol as a radiation sensitizer: a flow cytometric study. *Gynecol Oncol*, 50, 89-93 (1993)).

Like the Mi-2 β silencing by shMi-2 β (Supplementary Fig. 3C), we observed substantial increases in expressions of Mi-2 β target genes, Cxcl9 and Cxcl10, to similar levels after treatment of B16F10 cells with Z36-MP5 at concentrations 5 μ M to 100 μ M (Fig. 5F). Meanwhile, the increases in expression of both Cxcl9 and Cxcl10 were observed with an apparently dose-dependent fashion (Fig. 5F), indicating that 5 μ M to 100 μ M Z36-MP5 treatment would be sufficient for inhibiting Mi-2 β in cells. Moreover, we observed a maintenance of blood concentration of 4 μ M to 10 μ M Z36-MP5 in rat for around 1 hour after the intraperitoneal injection at a dose of Z36-MP5 as low as 1 mg per kg body weight (Supplementary Fig. 8C), suggesting the sufficient blood concentrations of Z36-MP5 in mice after intraperitoneal injection at a dose as high as 30 mg per kg body weight (estimated highest blood concentration in mice: around 0.97 mM) for Mi-2 β inhibition in vivo. Indeed, consistent with the cell assay data, treatment of B16F10-graft mice model with Z36-MP5 alone or in combination with anti-PD-1 perfectly recreated the in vivo data by shMi-2 β (Figs. 1 and 6), suggesting the Mi-2 β on-target inhibition in vivo.

We thank the review for the comment and understand the concern of the reviewer about the selectivity profiling of compound. To profile the selectivity of Z36-MP5 on ATPases in native cell lysates, we did our best to contact with all the commercially available services and chose the ActivX Biosciences Inc, as it provided the inhibitory profiling assay of compounds against the largest panel of 233 ATPases when preparing this manuscript. Unfortunately, the ATPases panel by ActivX Biosciences Inc does not include Mi-2 β protein. In this ATPase profiling assay, we showed that, against the panel of 233 diverse ATPases, Z36-MP5 possessed less than 35% inhibition at 1 μ M (Supplementary Table 3), a concentration that completely inhibits Mi-2 β ATPase. These results suggest that Z36-MP5 has a high Mi-2 β ATPase selectivity and specificity. Additionally, we observed the extremely high kinase selectivity of Z36-MP5 by KINOMEscan (Supplementary Fig. 7e), in which the inhibitory activity of Z36-MP5 against a panel of 468 diverse kinases were examined using an in vitro ATP-competitive binding assay.

To further identify the targeted specificity of Z36-MP5 in Mi-2 β , we generated a Mi-

2 β construct mutated at H727A and then tested the effect of Z36-MP5 by the FRET-based nucleosome repositioning assays. We found that Mi-2 β H727A mutant was no longer sensitive to Z36-MP5 (Supplementary Fig. 7f). This result further confirmed the specificity of Z36-MP5 to Mi-2 β protein.

Reviewer #3 (Remarks to the Author):

Overall, the authors have significantly revised their manuscript and answered most of the reviewer comments. But the clarity with regards to the mechanism behind immune sensitization of Mi-2B loss is still not sufficient and there are still three major points to be addressed before this paper is acceptable for publication:

1. While they show how Mi-2B loss activates the ISGs via EZH2 repression, they don't establish that this is the mechanism for the anti-tumor/ CD8 activation response. Akin to the figure S8f, they need to demonstrate that cells lacking EZH2 do not show a combinatorial effect when treated with anti-PD1 and Mi-2B inhibition.

Response: We thank the reviewer for the suggestion. We fully agree with the reviewer and performed the experimental therapeutics as suggested. Specifically, we implanted B16F10 cells into C57BL6 mice, treated the mice with MS177 to get the EZH2-targeted inhibition, and then gave anti-PD-1 in combination with Z36-MP5 or vehicles treatment. No treatment effects were detected in mice treated with MS177 to inhibit EZH2. This result further indicates the treatment effect of Z36-MP5 is EZH2 dependent (Fig. S8g in the revised manuscript).

2. All their flow data is still in percentage of CD4/CD8s and since the tumor sizes of the combination treatment in particular is much smaller, this is not an accurate representation. They need to show counts of these cells normalized by tumor weight, their counts from IHC alone is not sufficient to answer this critique.

Response: We thank the review for the comment and understand the concern of the reviewer. We have quantified the CD4/CD8 cells normalized by tumor weight as suggested. The cell numbers based on the result of flow cytometry analysis has been presented in Fig. 1e and Fig. 1c in the resubmitted manuscript).

3. The results with Z36-MP5 differ from their results using Mi-2B loss (shRNA or genetic loss)- the combinatorial effect with anti-PD1 is lower probably because there is no effect on CD4/T regs. The latter effect is consistently seen with both forms of genetic ablation of Mi-2B. They need to provide an explanation for this difference. For example, by comparing degree of EZH2 inhibition or activation of ISGs between the shRNA/genetic ablation of Mi-2B and the drug.

Response: We thank the reviewer for the great suggestion and has performed the experiment as suggested. Specifically, B16F10 cells with stable Mi-2 β silencing were

grafted into C57BL6 mice and then treated with Z36-MP5. The expression of H3K27me3, *Cxcl9* and *Cxcl10* were detected in mouse tumors. We found that the expression of H3K27me3, *Cxcl9* and *Cxcl10* were significantly repressed after Mi-2 β silencing or Z36-MP5 treatment. However, the Mi-2 β silencing-induced repression was more sufficient than Z36-MP5 treatment-induced repression (Please see the figure below the passage). Z36-MP5 is our first generation of Mi-2 β -targeted inhibitor. We will further improve its specificity and sensitivity by the structure modification.

Minor points:

-They refer to B16 implantation in B6 mice incorrectly as a xenograft model in a few places

Response: We thank the reviewer for pointing out these errors. We corrected the errors.

-Line 335 is confusing as it suggests that Mi-2B loss promotes T cell mediated cytotoxicity in vitro but their data doesn't support it and they reiterate in the rebuttals that Mi-2B loss doesn't alter efficacy of T cell killing.

Response: We thank the reviewer for pointing out this error. We apologize for this. We have corrected the description.

- Line 378 is only true with respect to B16 cells. In PM cells, even Z36-MP5 does increase H3K27ac levels, although the degree of increase may be less than that of GSK126. This requires a quantification and rewording on the statement to reflect the actual data.

"We confirmed these results, and found that Z36-MP5 repressed the level of H3K27me3, but did not activate H3K27ac"

Response: We thank the reviewer for the suggestion. We now correct it to: "We found that Z36-MP5 repressed the level of H3K27me3 in B16 and PM cells. However, compared to GSK126 treatment, the H3K27ac level only slightly increased in PM cells after treatment with Z36-MP5. "

REVIEWERS' COMMENTS

Reviewer #1 (Remarks to the Author):

The authors have fully and satisfactorily responded to all of my comments. Most importantly, they performed additional experiments to show enrichment of H3K27me3 by CHIP-qPCR at the Cxcl9 and Cxcl10 loci and performing TIMER analysis based on Cxcl9 and Cxcl10 expression levels from TCGA melanomas. They also corrected all editorial mistakes. This is a well-done study.

Reviewer #2 (Remarks to the Author):

Authors addressed all my concerns.

Reviewer #3 (Remarks to the Author):

I appreciate the efforts that authors have made to answer my comments and the ones of other reviewers. I have 2 more minor comments that should be addressed before publication

Comment to response 2: The y axis title of Figure 1e can be improved as it's slightly confusing (typically should read number of cells/mg or g tumor) and neither the legends nor the methods mentioned how the cell numbers of CD4/CD8 were normalized (to weight of tumor and volume of resuspension etc...).

This information should be added before before publication.

Comment on response 3: This comparison showing that the Z36-MP5 is not entirely as effective as complete genetic ablation of Mi-2B need to feature in some part of the manuscript as well, even if just as a sentence at the end of the results section or discussion (without the data as such if there is lack of figure space). Because as it stands the manuscript heavily emphasizes that Z36-MP5 is an "effective" inhibitor of Mi-2B without acknowledging that there is room for improvement to fully capture the immune advantages of complete Mi-2B silencing.

REVIEWERS' COMMENTS

Reviewer #1 (Remarks to the Author):

The authors have fully and satisfactorily responded to all of my comments. Most importantly, they performed additional experiments to show enrichment of H3K27me3 by ChIP-qPCR at the Cxcl9 and Cxcl10 loci and performing TIMER analysis based on Cxcl9 and Cxcl10 expression levels from TCGA melanomas. They also corrected all editorial mistakes. This is a well-done study.

Response: We sincerely appreciate your efforts of reviewing and the recognition of our improvement of our manuscript.

Reviewer #2 (Remarks to the Author):

Authors addressed all my concerns.

Response: We sincerely appreciate your efforts of reviewing and the recognition of our improvement of our manuscript.

Reviewer #3 (Remarks to the Author):

I appreciate the efforts that authors have made to answer my comments and the ones of other reviewers. I have 2 more minor comments that should be addressed before publication

Response: We sincerely appreciate your carefully review and your very helpful suggestions.

Comment to response 2: The y axis title of Figure 1e can be improved as it's slightly confusing (typically should read number of cells/mg or g tumor) and neither the legends nor the methods mentioned how the cell numbers of CD4/CD8 were normalized (to weight of tumor and volume of resuspension etc....).

This information should be added before before publication.

Response: We thank the reviewer for the constructive suggestion. The y axis title of Figure 1e has been replaced as suggested. Specifically, the y axis title of Figure 1e has been replaced: cell number / g tumor. We have also added the necessary information that how the cell numbers of CD4/CD8 were normalized (please see line 642-644 in the resubmitted manuscript).

Comment on response 3: This comparison showing that the Z36-MP5 is not entirely as effective as complete genetic ablation of Mi-2B need to feature in some part of the manuscript as well, even if just as a sentence at the end of the results section or discussion (without the data as such if there is lack of figure space). Because as it stands the manuscript heavily emphasizes that Z36-MP5 is an "effective" inhibitor of Mi-2B without acknowledging that there is room for improvement to fully capture the immune advantages of complete Mi-2B silencing.

Response: We thank the reviewer for the suggestion. According to suggestion, we have added the information in discussion. (please see line 445-447 in the resubmitted manuscript).